# *Staphylococcus aureus* thermonuclease NucA is a key virulence factor in septic arthritis
Ningna Li[1], Meghshree Vinod Deshmukh[2], Filiz Sahin[3], Nourhane Hafza[1], Aparna Viswanathan Ammanath [1], Sabrina Ehnert[3], Andreas Nüssler [3], Alexander N. R. Weber [4], Tao Jin [2] & Friedrich Götz [1] ✉

Septic arthritis, primarily caused by *Staphylococcus aureus*, poses a significant risk of both mortality and morbidity due to its aggressive nature. The *nuc1*-encoded thermonuclease NucA of *S. aureus* degrades extracellular DNA/RNA, allowing the pathogen to escape neutrophil extracellular traps (NETs) and maintain the infection unabated. Here we show that in the mouse model for hematogenous septic arthritis, the Δ*nuc1* mutant is much less pathogenic and the severity of clinical septic arthritis is markedly reduced, including decreased weight loss, lower kidney bacterial load, reduced bone erosion, and much less IL-6 production. In vitro, *S. aureus* genomic DNA induces a robust TNF-α response in macrophage-like RAW 264.7 cells abrogated when the DNA is degraded by NucA. Moreover, the wild type induces high levels of TNF-α, IL-10, and IL-6 in neutrophils and osteoblast-like SAOS-2 cells, respectively. NucA exacerbates septic arthritis by increasing extracellular and intracellular survival of bacteria.

Septic arthritis, the most aggressive joint disease carrying high mortality and morbidity risk, is predominantly caused by *Staphylococcus aureus*[1]. In half of the patients, even when they receive immediate treatment, the joint damage caused by septic arthritis is often irreversible, leading to permanent joint dysfunction[2]. It is known that innate immunity, including neutrophils and the complement system, protects from the development of septic arthritis[3,4].

*S. aureus* is one of the most successful bacterial pathogens because it expresses many colonization and pathogenicity factors, such as envelope-bound adhesins, or many secreted exoenzymes, including exotoxins, proteases, coagulase, collagenase, hyaluronidase, lipases, and nucleases. *S. aureus* encodes two nucleases: NucA, which is secreted and encoded by *nuc1*, and Nuc2, which is membrane-anchored with the C-terminus facing the extracellular environment and encoded by *nuc2*[5]. Of the two nucleases, it is the NucA that plays the crucial role in degrading extracellular DNA and RNA (eDNA and eRNA, respectively)[6,7].

The *nuc1* gene encodes a pre-pro-protein that is composed of a signal peptide, which is cleaved off by the signal peptidase, and a short pro-region which is processed by a protease releasing the mature and fully active NucA. NucA, also referred to as thermonuclease, is a $Ca^{2+}$-dependent, nonspecific endonuclease that catalyzes the hydrolysis of both DNA and RNA at the 5'

position of the phosphodiester bond-producing nucleoside 3'-phosphates and 3'-phosphooligonucleotide as end-products[8].

NucA functions to degrade extracellular DNA (eDNA), thus promoting the dispersal and destabilization of biofilm. Consequently, in the *S. aureus* Δ*nuc1* mutant biofilm formation was shown to be much more pronounced than in the parent strain. Additionally, neutrophils displayed higher NET formation when exposed to biofilms from the *nuc1* null mutant, and killed more bacterial cells, suggesting NucA is essential for the survival of *S. aureus* biofilms irrespective of whether the bacterium is phagocytosed or not[9,10]. Survival analysis in a hematogenous implant-associated infection mouse model indicated that *nuc1* expression is associated with higher mortality[11]. NucA also plays an important role in bacterial escape from NETs. NETs are released at sites of infection by activated neutrophils and consist of nuclear or mitochondrial DNA as a backbone with embedded antimicrobial peptides, histones, and cell-specific proteases, thereby providing an extracellular matrix to entrap and kill various microbes[12]. NucA delayed bacterial clearance in the lung and increased mortality after intranasal infection, thus promoting resistance against NET-mediated antimicrobial activity of neutrophils; consequently, the *nuc1* deficient mutant was significantly more susceptible to extracellular killing by activated

[1]Interfaculty Institute of Microbiology and Infection Medicine, University of Tübingen, Tübingen, Germany. [2]Department of Rheumatology and Inflammation Research, Institute of Medicine, Sahlgrenska Academy, University of Gothenburg, Gothenburg, Sweden. [3]Siegfried Weller Institute for trauma research, BG Unfallklinik Tübingen, University of Tübingen, Tübingen, Germany. [4]Interfaculty Institute for Cell Biology, Department of Immunology, Section Innate Immunity, University of Tübingen, Tübingen, Germany. ✉e-mail: friedrich.goetz@uni-tuebingen.de

neutrophils[13]. However, it is not only the degradation of DNA by NucA that is important for the escape from NETs but also the concomitant production of nucleoside 3'-phosphates and 3'-phosphooligonucleotides, which act as substrates for the adenosine synthase also secreted by *S. aureus*. The adenosine synthase converts the NucA products to deoxyadenosine, which triggers caspase-3-mediated apoptosis in immune cells[14].

Here we have investigated the role of NucA in a well-established mouse model of septic arthritis[15]. Our data show that in the hematogenous septic arthritis mouse model, NucA causes severe bone destruction, rapid weight loss, and high proinflammatory cytokine production. In vitro analysis suggests that reduced killing by neutrophils as well as the NET degrading activity of NucA and its induction of proinflammatory cytokines in certain host cells could be responsible for the high in vivo pathogenicity of a NucA-expressing strain.

## Results

### The *S. aureus* Newman Δ*nuc1* mutant is much less pathogenic in the mouse model of septic arthritis

To assess the role of NucA, *S. aureus* Newman wild-type (NWT) and its Δ*nuc1* mutant were evaluated in a mouse model of *S. aureus* septic arthritis. Naval Medical Research Institute (NMRI) mice were employed in this study. Mice received intravenous inoculations with either NWT or Δ*nuc1* and were observed for 7 days to monitor the infection process and immune response. Remarkably, Δ*nuc1*-infected mice exhibited minimal weight loss until day 7, whereas NWT-infected mice continued to lose weight up to 20% by the experiment's termination on day 7 (Fig. 1a). The severity of septic arthritis was assessed by clinical arthritis frequency and clinical arthritis

score (see "Materials and Methods" for details). Here, Δ*nuc1*-infected mice displayed significantly lower clinical arthritis symptoms than NWT-infected mice. Twenty percent of Δ*nuc1*-infected mice developed mild clinical arthritis symptoms up to day 7. In contrast, 40% of NWT-infected mice exhibited clinical arthritis symptoms as early as day 3, and by day 5, all animals had developed severe septic arthritis (Fig. 1b). Not only was the frequency of arthritis higher, but the clinical arthritis score was also elevated in mice infected with the NWT strain compared to the mutant strain (Fig. 1c). Importantly, both the kidney bacterial load (Fig. 1d) and kidney abscess scores (Fig. 1e) were significantly lower in the mice infected with the Δ*nuc1* mutant compared to those infected with its parental strain. Figure 1f illustrates the clear differences in kidney abscess formation between NWT- and Δ*nuc1*-infected mice.

To further confirm our clinical observations, we conducted micro-computed tomography (μCT) scans on all joints from mice inoculated with *S. aureus* (see Materials and Methods for details). Intravenous injection of *S. aureus* NWT into NMRI mice resulted in severe bone destruction in 12% of joints after 7 days post-infection, whereas mice infected with the Δ*nuc1* mutant showed almost no sign of bone erosion (Fig. 2a and b). Figure 2c shows representative 3D images of a wrist, a knee, and a shoulder from mice infected either with the Δ*nuc1* mutant or the NWT strain.

### Infection with wild-type *S. aureus* results in a significant increase in the levels of IL-6 and S100A8/A9

IL-6 and TNF-α are essential for septic arthritis development[16,17]. S100A8/A9 serves as a predictor of septic arthritis in bacteremic mice, and KC (CXCL1)

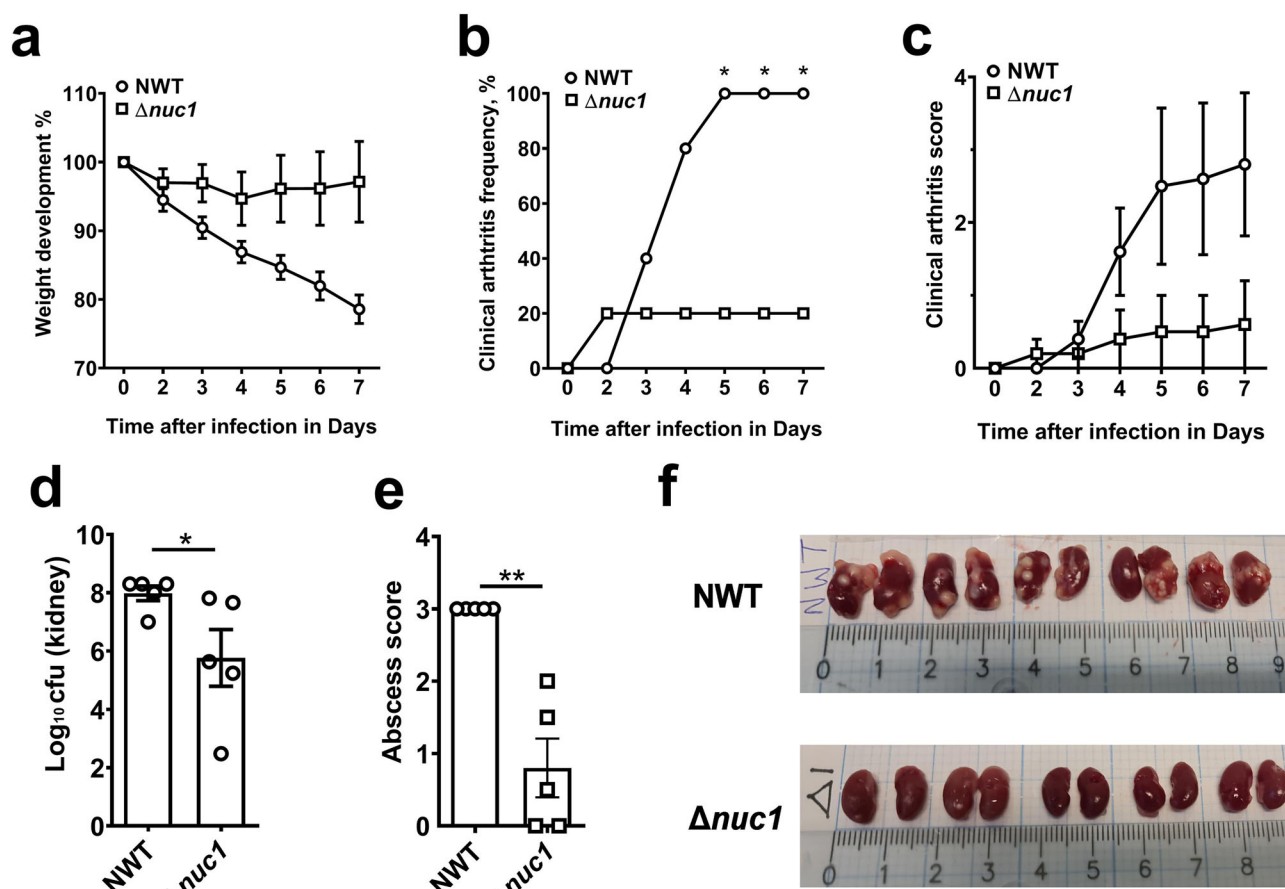

**Fig. 1 | *S. aureus* Newman wild-type strain (NWT) imparts more severe arthritis and virulence than its Δ*nuc1* mutant during infection in NMRI mice. a** Weight development, (**b**) clinical arthritis frequency, and (**c**) score were measured from mice infected with either NWT or Δ*nuc1* mutant for 7 days. **d** Bacterial counts in the kidney, (**e**) kidney abscess score, and (**f**) representative kidney abscess images from mice were investigated on day 7 post-infection. Statistical analyses were performed using the Mann-Whitney U test (**a**, **c**), where the data were presented as the mean ± SEM (standard error of the mean). Statistical significance: not significant, $p > 0.05$; *$p < 0.05$; **$p < 0.01$.

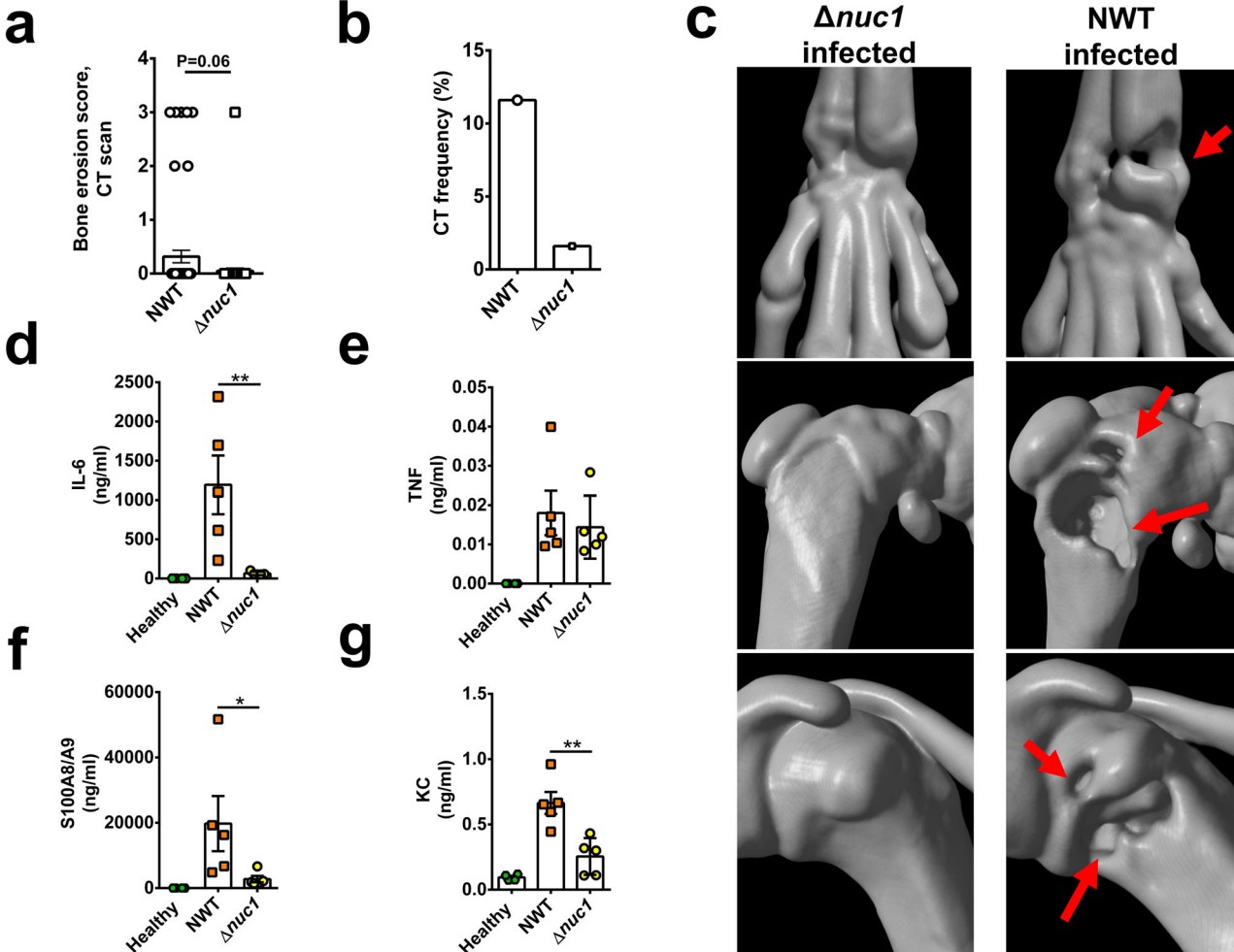

**Fig. 2 | Monitoring bone erosion by microcomputed tomography (µCT) and cytokine levels in mice infected with NWT or Δnuc1. a** Bone erosion score and (**b**) CT frequency of the joints in NMRI mice intravenously injected with *S. aureus* NWT or its Δnuc1 mutant on day 7 post-infection. **c** Representative 3D images of micro-computed tomography (µCT) scanning of the mice joints (hand, knee, shoulder) created with µCT scanner after infection. Right panel: NWT; left panel: Δnuc1; red arrow indicates bone erosion. **d–g** Levels of IL-6, TNF-α, KC, and S100A8/A9 were measured in plasma derived from NMRI mice intravenously injected with NWT or Δnuc1. Statistical analyses were performed using the Fisher exact test (**a**) and Mann-Whitney U test (**b**, **d–g**), where the data were represented in mean ± SEM (**b**, **d–g**)., where the data were represented in mean ± SEM. Statistical significance: not significant, *p* > 0.05; \**p* < 0.05; \*\**p* < 0.01.

recruits neutrophils, key innate immune cells essential for disease control[4,18]. Accordingly, on day 7 post-infection we collected blood samples from the infected mice and measured the levels of these immune mediators: IL-6, TNF-α, KC (CXCL1), and S100A8/A9. As shown in Fig. 2d-g, the levels of IL-6, KC (CXCL1), and S100A8/A9 were markedly reduced in Δnuc1-infected mice as compared to NWT-infected mice. While the TNF-α levels tended to be lower in Δnuc1-infected mice, the difference did not reach statistical significance. Collectively, our results indicate that NucA is a crucial virulence factor for *S. aureus* pathogenicity in an infection model of septic arthritis. We further investigated possible reasons for the reduced pathogenicity of the Δnuc1 mutant in the septic mouse model using different host cells.

### NucA digestion of gDNA decreases TNF-α production in mouse macrophages

*S. aureus* produces various virulence factors and triggers inflammation. Previous research found that the DNA from *S. aureus*, containing unmethylated CpG motifs, acts as a factor that triggers arthritis and septic shock[19,20]. Unmethylated bacterial DNA and CpG motifs are recognized by the toll-like receptor 9 (TLR9), which is expressed in immune cells such as macrophages and dendritic cells[21,22]. Recognition via the TLR9 pathway initiates the host response to *S. aureus* infection.

As staphylococcal macromolecules are frequently contaminated with lipoproteins/lipopeptides that are sensitively detected by Toll-like receptor 2 (TLR2) at picomolar levels, cytokine induction could be due to these constituents. Therefore, the mutant *S. aureus* USA300 LAC JE2Δlgt was tested here as a control to recognize a possible interference of TLR2 and TLR9 ligands on cytokine production. JE2 is a plasmid-cured derivative of USA300 LAC strain, which is still methicillin-resistant and an important model strain to study *S. aureus* virulence[23]. JE2Δlgt lacks the phosphatidylglycerol: prolipoprotein diacylglyceryl transferase Lgt, and therefore no lipidation of lipoproteins takes place and no TLR2 response can be triggered by this mutant[24,25]. By including this mutant and the double mutant JE2Δnuc1Δlgt together with JE2 and JE2Δnuc1 in the comparative immunostimulation studies, it is possible to specifically detect NucA-induced cytokine induction.

The mouse macrophage-like RAW 264.7 cells were treated with increasing concentrations of gDNA from JE2Δlgt and RAW 264.7 (negative control). The standard CpG oligonucleotides ODN2006 and the non-CpG oligonucleotides ODN2137 were used as positive and negative controls, respectively (Fig. 3a). gDNA from *S. aureus* JE2Δlgt increased the production of the proinflammatory cytokine TNF-α in mouse macrophage-like RAW 264.7 cells in a concentration-dependent manner, while gDNA from RAW 264.7 cells did not (Fig. 3a). At a dose of 10 ng/ml, *S. aureus* gDNA

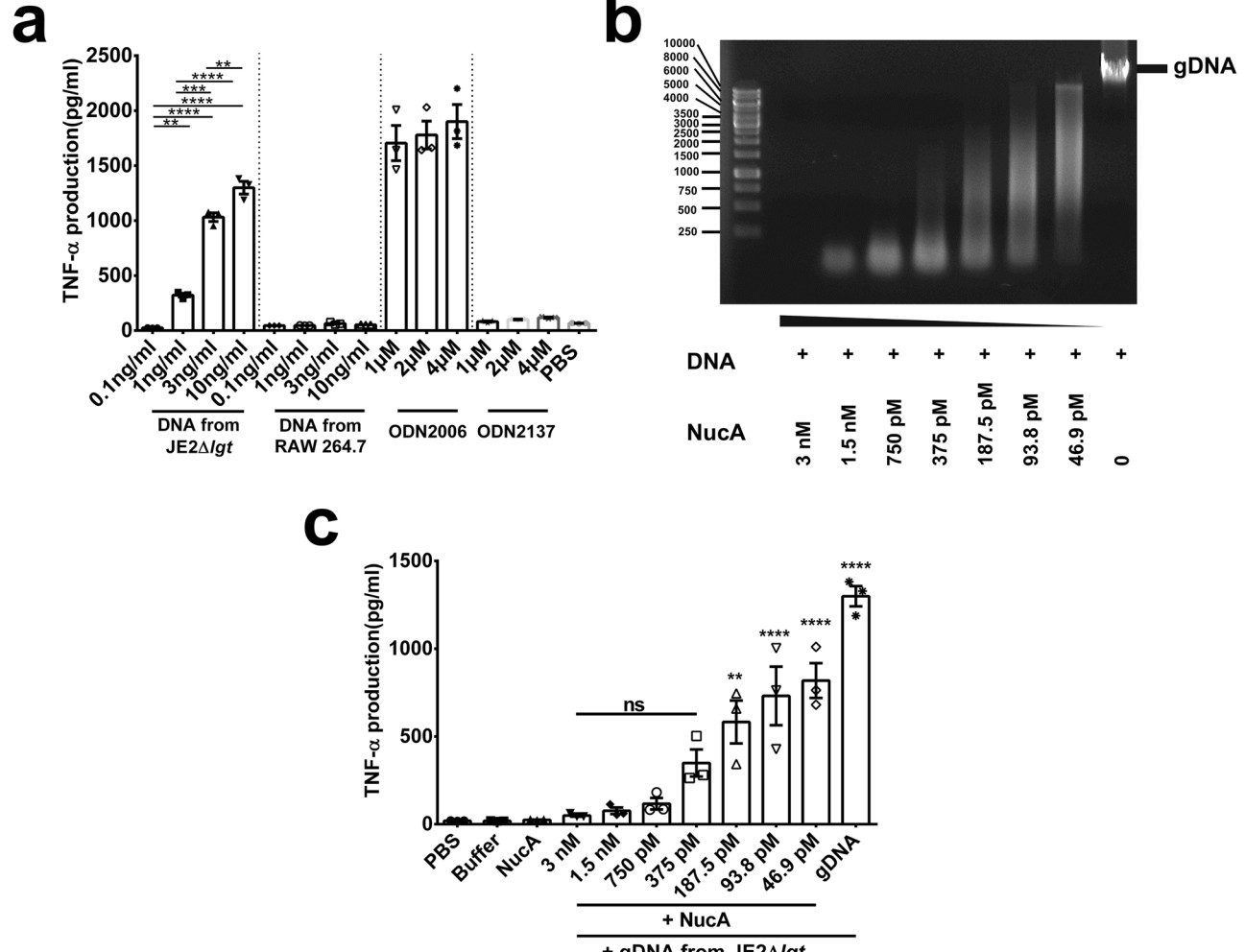

**Fig. 3 | Induction of TNF-α by macrophages upon exposure to diverse DNA.**
**a** Stimulation of mouse macrophage RAW 264.7 cells with JE2Δ*lgt* gDNA, mammalian gDNA, ODN2006 (CpG oligonucleotide), and ODN2137 (GpC dinucleotides) at varying concentrations for 18 h, followed by TNF-α measurements.
**b** Agarose gel analysis showed the degradation of JE2Δ*lgt* gDNA by gradually decreasing NucA concentration. **c** Undigested and digested JE2Δ*lgt* gDNA were incubated with RAW 264.7 cells for 18 h. Released TNF-α in cellular supernatants was quantified by ELISA. Data represent the mean ± SEM from three independent experiments; ns(not significant), $p > 0.05$; *$p < 0.05$; ****$p < 0.0001$, one-way ANOVA with Dunnett's posttest.

induced the release of high TNF-α levels, as did exposure to 1 μM ODN2006, indicating a high immunostimulatory activity of the staphylococcal DNA.

Comparative virulence studies in different *S. aureus* model strains have shown that the USA300 and Newman strain not only have comparable lethality in a mouse sepsis model but are also similar in their hemolytic activity and biofilm formation[26]. However, in other studies have shown that the two strains exhibit distinct virulence properties; their pathogenicity differs depending on the infection model[27–29]. Both strains also express NucA and the corresponding Δ*nuc1* mutants could be complemented by pRB473-*nuc1* (Supplementary Fig. 1a). Therefore we used both for the whole study. In *S. aureus*, NucA is secreted into the supernatant and degrades extracellular DNA and RNA[5]. This was demonstrated when we incubated the supernatant of JE2 and its mutants JE2Δ*nuc1*, JE2Δ*lgt*, and JE2Δ*nuc1*Δ*lgt* with *S. aureus* gDNA for 1 h to evaluate the nuclease activity. In all mutants in which *nuc1* was deleted, the gDNA remained intact, whereas in JE2 and the JE2Δ*lgt* mutant, the gDNA was completely degraded, indicating that NucA is responsible for gDNA degradation (Supplementary Fig. 1b, 7b). Furthermore, the degradation of eDNA by recombinant NucA, which was expressed and purified from *Escherichia coli* (Supplementary Figs. 2, 7c), suggested that NucA plays a role in controlling DNA/RNA-dependent immune stimulation. Indeed, the stepwise degradation of *S. aureus* gDNA by increasing amounts of NucA (from 46 pM to 3 nM)

correlated with a stepwise decrease in TNF-α production (Fig. 3b, c and Supplementary Fig. 7a).

## The *S. aureus nuc1* deletion mutants have an impact on cytokine production in various cell lines

*S. aureus* activates different host cells, including macrophages, neutrophils, and osteoblasts[30–32]. We further investigated whether the immune stimulation of JE2 and its mutants differed in RAW 264.7, neutrophils, and osteoblast-like SAOS-2 cells. We incubated live *S. aureus* JE2 and its JE2Δ*nuc1*, JE2Δ*lgt*, and JE2Δ*nuc1*Δ*lgt* mutants for 18 h with RAW 264.7, SAOS-2 cells, and 5 h with neutrophils, respectively. To assess the immune responses elicited by different bacterial strains, we measured cytokine production, including classically pro-inflammatory mediators such as TNF-α and IL-6, as well as anti-inflammatory cytokines such as IL-10 and IL-1Ra. IL-10 was also very high in sepsis patients, and IL-1Ra and IL-10 also influenced the inflammatory response[33–35]. There was no difference in IL-6 and TNF-α production in RAW 264.7 upon exposure to either JE2 or JE2Δ*nuc1* (Fig. 4a). In SAOS-2 cells, IL-6 production was decreased in response to all mutants, but there was no difference in TNF-α production. In neutrophils, the production of TNF-α, IL-10, and IL-1Ra was markedly decreased in response to all the mutants (Fig. 4b, c), particularly those lacking *lgt* (JE2Δ*lgt* and JE2Δ*nuc1*Δ*lgt*). It is not unexpected that the Δ*lgt*

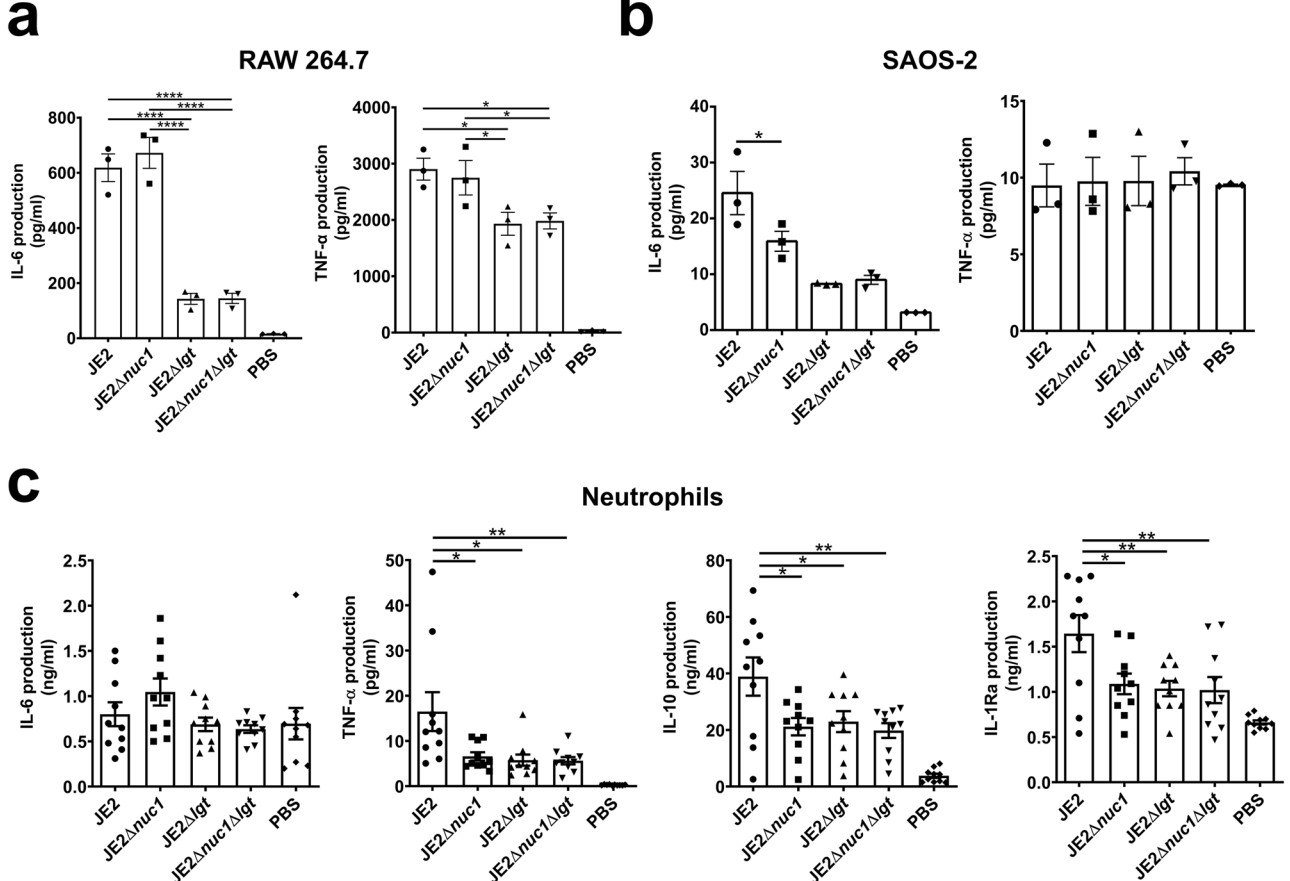

**Fig. 4 | Induction of cytokines by various host cells upon exposure to live JE2, JE2Δ*nuc1*, JE2Δ*lgt*, and JE2Δ*nuc1*Δ*lgt*.** The PBS-washed bacteria were incubated with (**a**) RAW 264.7 cells at an MOI = 30, (**b**) SAOS-2 cells at an MOI = 3, and (**c**) with neutrophils at an MOI = 50. Cellular supernatants were collected after 18 h for RAW 264.7 and SAOS-2 cells, and after 5 h for neutrophils to measure various cytokines by ELISA assay. For neutrophils, the experiments were displayed from $n = 10$ donors. Triplet experiments were conducted; error bars indicate ± SEM; not significant, $p > 0.05$; *$p < 0.05$; **$p < 0.01$; and ****$p < 0.0001$, one-way ANOVA with Dunnett's posttest.

mutants induce less cytokines since lipoproteins/lipopeptides trigger a very strong innate immune response. As compared with NWT, there was less IL-6 production in RAW 264.7 and no significant difference in IL-6 and TNF-α in SAOS-2 when exposed to the Δ*nuc1* strain (Supplementary Fig. 3). As the generated mutants showed hardly any difference in growth and hemolytic activity as compared to the parental strain (Supplementary Fig. 1a, c and d), we assume that all the effects seen are mainly due to the deletion of *nuc1* or *lgt* genes.

### JE2Δ*nuc1* exhibits decreased internalization or survival in various host cells

*S. aureus* can be engulfed by multiple cells[36,37]. We determined bacterial survival and phagocytosis in neutrophils, following a previously used protocol[38,39]. With different conditions tested, we found that better results were obtained with an MOI (multiplicity of infection)=2 for the phagocytosis assay and an MOI = 0.1 for the bacterial killing assay. Under these conditions, the survival of JE2Δ*nuc1* was decreased already at early time points as compared to JE2 (Fig. 5a). This was consistent with a lower phagocytosis index in JE2Δ*nuc1* (Fig. 5a). This led us to investigate whether live JE2 and its mutants differ in internalization by other host cells such as RAW 264.7. To assess only intracellular survival, membrane adherent and extracellular bacteria were killed after 1.5 h incubation, and then the CFU of internalized bacteria per host cell was determined. In RAW 264.7 we observed lower intracellular numbers of JE2Δ*nuc1* cells as compared to JE2 and the complementation strain JE2Δ*nuc1*(pRB473-*nuc1*), suggesting that survival was affected by *nuc1* (Fig. 5b).

### Effect of live bacteria and supernatant on NETs formation and clearing

Neutrophils act as the first defense line of the innate immune system and form NETs to clear pathogens[38]. NETs consist of a DNA backbone coated with various proteins, such as myeloperoxidase (MPO), nuclear proteins (histones), neutrophil elastase (NE), and calprotectin[40,41]. In *S. aureus*, NucA can degrade extracellular DNA, thereby reducing NET formation and evading immune clearance. Here, the neutrophils were exposed to live bacteria (MOI = 1:2) and bacterial supernatants (2%) for 2 h following eDNA-staining with SYTOX Green. Live bacteria caused a slow increase in NETosis, whereas the bacterial supernatant induced a stronger NET release in a shorter time (Fig. 6a, b). Additionally, neutrophils were still viable after 3 h incubation (Supplementary Fig. 4).

Immunofluorescence staining of NET formation (Fig. 6c and Supplementary Figs. 5, 6) further confirmed NETosis occurred, as visible by positive staining for DNA (blue), MPO (green), and citrullinated histone H3 (CitH3, red) after 1 h incubation of neutrophils with bacterial cell-free supernatant but not with live bacteria. NET was higher in response to JE2Δ*nuc1* mutants and their supernatants than in the parental strain, as also can be observed in fluorescence microscopy images (Fig. 6c and Supplementary Fig. 6). This suggests that the supernatant of Δ*nuc1* mutants is less effective in degrading DNA, resulting in increased levels of NETs.

### Discussion

The comparative studies between *S. aureus* parental strain and the Δ*nuc1* mutant revealed that NucA is an important virulence factor in septic

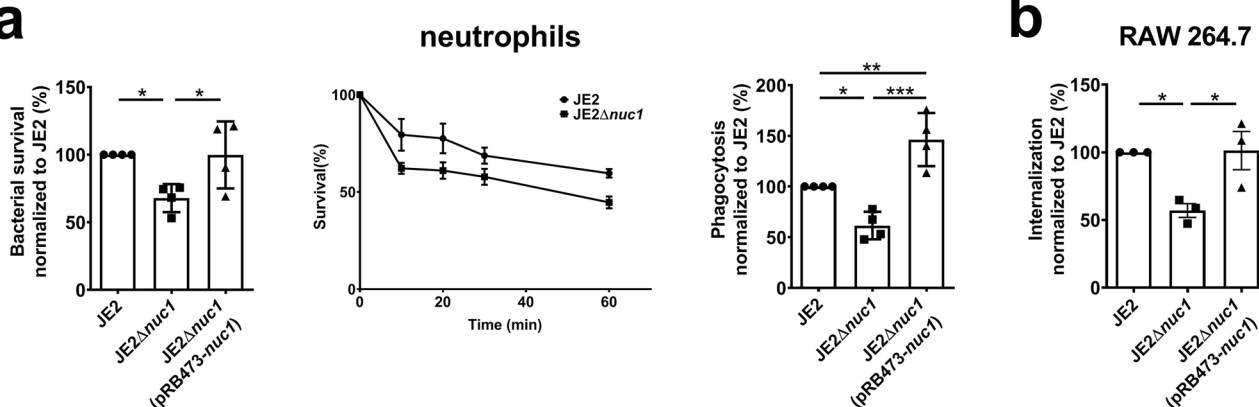

**Fig. 5 | Bacterial killing and phagocytosis studies in various cells. a** For bacterial killing in neutrophils, JE2, JE2Δ*nuc1* and JE2Δ*nuc1* complemented with plasmid-expressed *nuc1* (JE2Δ*nuc1* pRB473-*nuc1*) were opsonized with 10% human pooled serum. Neutrophils were incubated with bacteria at an MOI = 0.1 for 1 h and bacterial survival was checked at 10 min, 20 min, 30 min, and 1 h after exposure to neutrophils. Fluorescence-labeled bacteria were incubated with neutrophils at an MOI of 1:2 and the phagocytosis index was calculated 1 h after incubation via FACS. **b** To investigate bacterial survival in RAW 264.7, RAW 264.7 was incubated with

JE2, JE2Δ*nuc1,* and JE2Δ*nuc1*(pRB473-*nuc1*) at an MOI of 1:20 for 1.5 h. Extracellular and attached bacteria were removed by gentamicin and lysostaphin before lysing the cells. Neutrophils used in these experiments were obtained from 4 donors and the other experiments were performed at least three times; error bars indicate mean ± SEM. Statistical analyses were performed using one-way ANOVA with Dunnett's posttest. Statistical significance: not significant, $p > 0.05$, *$p < 0.05$; **$p < 0.01$; and ***$p < 0.001$.

arthritis. *S. aureus* NWT-infected mice showed marked weight loss, much increased clinical arthritis frequency, a 3-fold higher abscess score, severe bone erosion, and very high IL-6 content in the plasma. In contrast, the Δ*nuc1*-infected mice showed hardly any signs of septic arthritis, almost no weight loss, their clinical arthritis and abscess scores were much lower, and the bacterial load in kidneys was decreased (Fig. 1). Most remarkable, however, was that Δ*nuc1*-infected mice showed almost no bone erosion in the joints (Fig. 2a, b, c). Here, we provide three lines of in vitro evidence to support our in vivo observations: 1) NucA efficiently degrades bacterial DNA, which can trigger the release of proinflammatory cytokines like TNF-α, known for their critical roles in septic arthritis development; 2) NucA production may increase the intracellular survival of *S. aureus*; 3) NucA produced by *S. aureus* effectively digests NETs formed by neutrophils, potentially aiding bacterial evasion from innate immune killing, and thereby promoting bacterial survival and exacerbating disease severity.

Most bacteria release DNA and RNA during proliferation. In *S. aureus* DNA is released during cell lysis resulting from induction of prophages or activation of proteins with holin-like properties such as CidA and LrgA[42]. The secreted NucA degrades eDNA/RNA very efficiently to allow the reuse of the degradation products. How powerful NucA is in degrading eDNA is illustrated in Supplementary Fig. 1b and 7b. When the supernatants of JE2 and its mutants were incubated with gDNA, it was completely degraded within 1 h by the supernatant of JE2 but not by the Δ*nuc1* mutants. This efficient degradation of eDNA not only ensures the reuse of nucleic acid building blocks but also the escape of staphylococci from a biofilm community or NETs, providing NucA-expressing bacteria a clear advantage during infection[43].

It is known that bacterial DNA triggers an inflammatory response and induces cytokine production via the TLR9 receptor[21,22]. Further studies in mice have proven that DNA from *S. aureus* results in arthritis[20,44]. To check the potential involvement of bacterial DNA and the role of NucA in inflammation, gDNA was isolated from the JE2Δ*lgt* mutant to avoid lipoprotein contamination. The NucA-digested DNA and intact DNA were then incubated with RAW 264.7 cells. In these cells, staphylococcal DNA induced TNF-α production in a dose-dependent manner (Fig. 3a); as the gDNA was progressively degraded by increasing concentrations of NucA, TNF-α production decreased with progressing degradation (Fig. 3b, c and Supplementary Fig. 7a). Similar results were also obtained with Group A *Streptococcus* (GAS), which produces the DNase Sda1 to prevent IFN-α and TNF-α secretion by murine macrophages[45]. TNF-α is recognized as

pathogenic in the initiation and progression of septic arthritis[16]. Moreover, combining antibiotics with a TNF-α inhibitor yielded superior results as compared to antibiotics alone, effectively reducing synovitis and joint destruction in a mouse model of septic arthritis[46]. Simultaneously, TNF-α plays a crucial role in the Th1 response and primes phagocytes for effective elimination of pathogens[47]. Anti-TNF-α treatment has been shown to compromise immune killing efficacy, leading to increased kidney bacterial load in a mouse model of *S. aureus* septic arthritis[48]. Hence, NucA disrupts the immune-stimulating effect of bacterial DNA, potentially leading to an elevated kidney bacterial load.

As we showed that NucA-digested bacterial eDNA has no immune-stimulating activity in murine macrophages, we expected that the Δ*nuc1* mutant, in which eDNA is not degraded (Supplementary Fig. 1b, 7b), would elicit a stronger immune response than the parental strain in various cell types. *S. aureus* stimulates the immune response in various cells, including macrophages and neutrophils, as well as non-immune cells, such as osteoblast cells[30–32]. In addition, osteoblasts and osteoclasts are responsible for bone remodeling and construction[49,50]. Accordingly, macrophages RAW 264.7, osteoblast cells SAOS-2 and neutrophils were employed in this study. JE2Δ*nuc1* mutant triggered no IL-6 or TNF-α production in RAW 264.7 (Fig. 4a, b). What we see is that lipoproteins play a decisive role in immune stimulation in RAW 264.7 cells, which is in agreement with earlier results[25,51,52]. However, JE2Δ*nuc1* mutant induced less IL-6 in SAOS-2 cells and less TNF-α in neutrophils, while NWTΔ*nuc1* induced less IL-6 in RAW264.7 cells (Fig. 4 and Supplementary Fig. 3). In addition to measuring pro-inflammatory cytokines, we also measured the production of anti-inflammatory cytokines, IL-10 and IL-1Ra, in neutrophils to assess the balance between pro- and anti-inflammatory responses during infection. IL-10 is critical for regulating and balancing the infection and immune response, which typically increases during inflammation[53,54]. IL-1Ra, as a natural antagonist of IL-1 signaling and related to extensive IL-10 production, could influence the inflammatory response and is used to treat inflammation[55,56]. IL-1R knock-out mice developed more severe septic arthritis, but IL-1R treatment for inflammation-related diseases could also increase the infections in sepsis[35,48,57]. When neutrophils were exposed to the parental strain and its mutants, JE2Δ*nuc1* showed less IL-10 and IL-1Ra production. Those results give a hint that NucA has an impact on immune stimulation.

The next key question that arises in this context is the molecular causes of NucA-induced septic arthritis and bone erosion. To investigate this, we

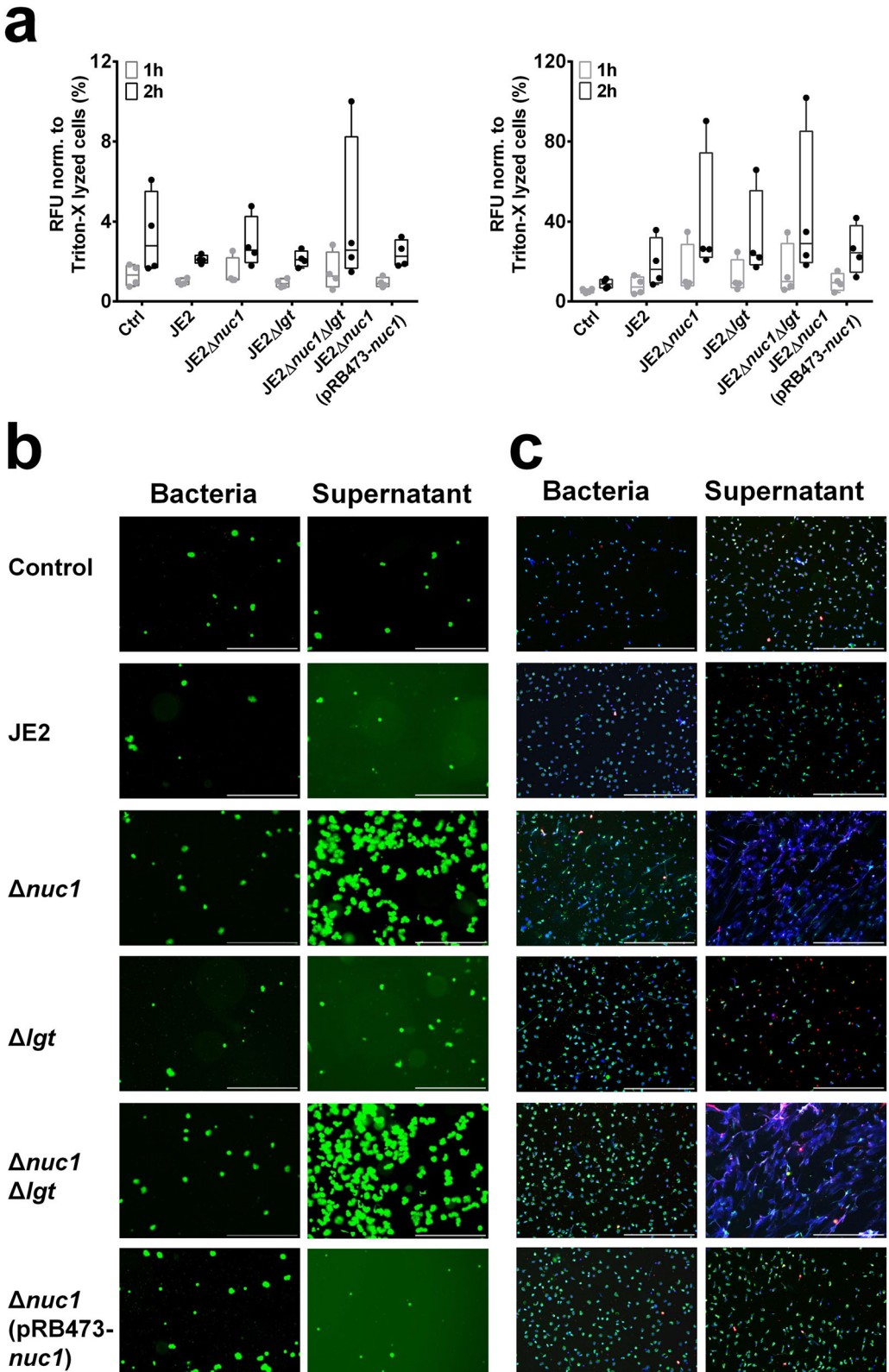

**Fig. 6 | NET formation upon exposure to live bacteria and supernatants of JE2 and its mutants. a** Sytox Green assay of neutrophils exposed to JE2, JE2Δ*nuc1*, JE2Δ*lgt*, and JE2Δ*nuc1*Δ*lgt* with live bacteria (MOI = 1:2) or overnight supernatant (2% volume) for 2 h. Relative fluorescence units (RFU) of Sytox Green normalized to Triton X-100-lysed neutrophils are shown. **b** Representative fluorescence images of Sytox Green staining after 3 h of incubation. Scale bar = 500 μm. **c** Representative images of immunofluorescent staining after 1 h of incubation. Blue: DNA (Hoechst 33342); Green: myeloperoxidase, MPO; Red: citrullinated histone H3, citH3, scale bar: 500 μm. The graph displays the average values ± SEM obtained from 4 donors; not significant, *p* > 0.05, one-way between-groups ANOVA with Dunnett's posttest.

measured the levels of several cytokines in the blood of mice on 7 days post-infection. Previous studies have shown that IL-6 and TNF-α are critical for septic arthritis development[16,17]. Patients who have septic arthritis exhibit high concentrations of TNF[58,59]. Lack of TNF-α has been associated with an inefficient ability to clear bacteria[16]. In addition, KC attracts neutrophils that control septic arthritis, and S100A8/A9 has been identified as an early predictor of septic arthritis during *S. aureus* bacteremia[4,18]. Since in vivo experiments showed minimal differences in TNF-α induction after infection with WT or *nuc1* mutant, we further investigated this phenotype in vitro. However, in macrophage-like and osteoblast-like cells, JE2 and JE2Δ*nuc1* did not cause any significant difference in TNF-α production (Fig. 4a, b and Supplementary Fig. 3). In neutrophils, more TNF-α production was induced by the parental strain than by the *nuc1* mutant. In the septic arthritis mouse model, NWT induced an almost 100-fold higher IL-6 production in plasma than its Δ*nuc1* mutant. The very high IL-6 content in the blood reflects the severity of septic arthritis by infection with the NucA-expressing strain. In fact, sepsis or acute respiratory distress syndrome is correlated with an increased IL-6 content. However, there is no significant difference in IL-6 production between JE2 and its Δ*nuc1* mutant in the different cell cultures, showing once again that the in vitro situation does not always reflect the in vivo situation. IL-6, which is mainly produced by macrophages and T lymphocytes in response to pathogens, is not only a key player in rheumatoid arthritis, but it also promotes megakaryocyte maturation and the release of platelets when reaching the bone marrow. IL-6 has emerged alongside IL-1 and TNF as a master regulator of inflammation: it is essential for innate and adaptive immunity, is required for efficient pathogen clearance, and has important physiological roles in humans regulating the acute-phase response, hematopoiesis, metabolic rate, lipid homeostasis, and neural development.

The pronounced upregulation of S100A8/A9 in NWT-infected mice compared to the Δ*nuc1*-infected mice could be induced by IL-6-triggered inflammation and leukocyte recruitment[60,61]. The high levels of IL-6 and S100A8/A9 at the end of the mouse experiment reflect the severity of infection. Inflammatory cytokines play a role in the bone remodeling process. For example, in IL-6 deficient mice, bone erosion was reduced[62]. Therefore, the NucA-expressing strain could cause high IL-6 production, which may lead to uncontrolled progression of bone destruction by osteoclasts.

Furthermore, we also found that the JE2Δ*nuc1* mutant exhibited reduced survival, which could explain why we detected less bacteria inside neutrophils 1 h after exposure to neutrophils (Fig. 5a). This encouraged us to investigate the internalization in other cell lines. As shown in Fig. 5b, we also observed reduced numbers of JE2Δ*nuc1* in RAW 264.7 cells. These findings indicate that NucA increases *S. aureus* survival, likely both extracellularly and intracellularly, which may contribute to an increase in the severity of septic arthritis.

In hematogenous septic arthritis, compromised innate immune defenses increase the likelihood that bacteria in the bloodstream invade joints, ultimately leading to the development of septic arthritis[15]. Neutrophils are recognized as vital immune cells guarding against *S. aureus* septic arthritis. Neutrophil-depleted mice exhibited heightened and more frequent septic arthritis, coupled with compromised bacterial clearance as evidenced by elevated CFU counts in both blood and kidneys[4]. We also compared the effects of live bacteria and the corresponding culture supernatant on degradation of neutrophil extracellular traps (NETs). NETs represent a form of innate immune response that prevents microorganisms from spreading, while the high local concentration of antimicrobial agents may kill bacteria[63,64]. The *S. aureus* thermonuclease NucA is known to degrade the DNA within NETs to escape scavenging and killing[38]. It has been shown that the combined activity of *S. aureus* nuclease and adenosine synthase converts extracellular DNA to deoxyadenosine (dAdo) in staphylococcal abscesses. Human equilibrative nucleoside transporter 1 (hENT1) mediates dAdo transportation in macrophages, leading to caspase-3-induced apoptosis[14,65,66]. This could be happening in our model as well, but we did not analyze the viability of macrophages isolated from abscesses.

When we exposed neutrophils to live *S. aureus*, we observed a time-dependent increase in stained eDNA, but there was no pronounced difference between stimulation with JE2 and its mutants (Fig. 6a, b) until 2 h after exposure. When we incubated neutrophils with the corresponding supernatants, we observed a clear difference in the response to JE2Δ*nuc1* mutants as compared to other strains (Fig. 6c and Supplementary Fig. 6). Immunofluorescence revealed a stronger DNA signal in response to JE2Δ*nuc1* supernatants, suggesting decreased NET degradation. The question remains as to why we did not see a major difference between the wild-type and the Δ*nuc1* live bacteria. We assume that by washing the bacterial cells with PBS, NucA is washed out and the bacteria are rapidly phagocytosed and killed when incubated with neutrophils.

Overall, our study suggests that NucA plays a crucial role in the pathogenesis of *S. aureus* septic arthritis, as evidenced by Δ*nuc1*-infected mice showing reduced arthritis severity, bone erosion, and kidney abscess formation, alongside lower bacterial loads. In vitro data further support these findings, demonstrating that NucA degrades bacterial DNA, shields *S. aureus* from killing, and digests neutrophil extracellular traps, ultimately promoting bacterial survival and worsening disease severity. How NucA triggers severe bone erosion is the subject of further research. In a summary image, we illustrated the main differences between NucA-producing and -non-producing *S. aureus* strains (Supplementary Fig. 8).

## Materials and methods
### Bacterial strains, plasmids, primers, and culture conditions
Bacterial strains and plasmids used in this study are described in Supplementary Table 1. All the primers are listed in Supplementary Table 2. *Escherichia coli* BL21 (DE3) was grown in Luria-broth medium (LB), and *Staphylococcus aureus* strains were cultured in tryptic soy broth (TSB, Millipore, Merck) or basic medium (BM) broth (Luria broth supplemented with 0.1% $K_2HPO_4$ and 0.1% glucose) or stored as previously mentioned[67]. To keep the plasmids in the bacteria, *E. coli* was supplied with ampicillin or kanamycin and *S. aureus* was supplied with 10 μg/ml chloramphenicol. For comparison of growth kinetics, the *S. aureus* JE2, Newman, and their mutants were grown in TSB medium, and measured with Varioskan LUX Multimode Microplate Reader (Thermo Fisher) for 24 h.

### Deletion of *nuc1* and *lgt* in *S. aureus*
*nuc1* is the nuclease1-encoding gene and *lgt* is the lipoprotein diacylglyceryl transferase enzyme-encoding gene[7,25]. For deleting *nuc1* and *lgt* genes, the knockout plasmid pBASE6 was employed. The disruption primers were designed to contain upstream and downstream of the target gene regions. Fragments were generated by PCR and subcloned into EcoRV-digested pBASE6 vector, resulting in pBASE6Δ*nuc1* and pBASE6Δ*lgt*. The resulting plasmids were transformed into *E. coli* DC10B for amplification. Then, plasmids were isolated and verified by DNA sequencing. The correct plasmids pBASE6Δ*nuc1* and pBASE6Δ*lgt* were transformed into an intermediate host, *S. aureus* RN4220, by electroporation to restrict foreign DNA and then into *S. aureus* JE2 or *S. aureus* Newman (NWT). The process for deletion of *lgt* and *nuc1* from *S. aureus* was followed as previously described[68]. Positive colonies were incubated in TSB with 10 μg/ml chloramphenicol (Cm) at 43 °C overnight and then transferred to TSB supplemented 7.5 μg/ml Cm at 43 °C for another overnight incubation. The culture was plated and a single colony was picked for inoculation at 30 °C. The overnight culture was diluted and plated onto TSA plates containing 1 μg/ml anhydrotetracycline (ATc) for two days. Colonies from the plate were streaked on TSA with or without Cm. The colonies that could grow on TSA plate but not on TSA with Cm were selected for PCR verification, resulting in JE2Δ*nuc1*, NewmanΔ*nuc1*, JE2Δ*lgt*, and JE2Δ*nuc1*Δ*lgt*.

### Construction of complementation strain
For complementation, the plasmid pRB473-*nuc1* was introduced. The *nuc1* fragment was amplified, ligated to pRB473 plasmids, and transformed into *E. coli* DC10B. Positive colonies were selected and verified via DNA sequencing. The correct plasmid was purified and transformed

into JE2$\Delta nuc1$ and $\Delta nuc1$, yielding JE2$\Delta nuc1$(pRB473-*nuc1*) and $\Delta nuc1$(pRB473-*nuc1*).

### Hemolytic activity and nuclease activity assays
Bacteria were grown in TSB medium overnight. The OD578 of overnight culture was adjusted and spotted on the Blood Agar (TSA with Sheep Blood) plates (Thermo Fisher) at 37 °C, and then the hemolysis zone was measured. The supernatants of overnight culture were checked for nuclease activity using DNase Test Agar with Toluidine Blue (Merck, Millipore). The DNase Test Agar plates were incubated at 37 °C.

### Mouse model for *S. aureus* septic arthritis
To compare the pathogenicity of *S. aureus* Newman wild-type strain (NWT) and its $\Delta nuc1$ mutant, a septic arthritis mouse model was used. NMRI female mice, aged 8 weeks, were purchased from Envigo (Venray, Netherlands). All mice were housed at the animal facility at the University of Gothenburg. Mice were kept under standard temperature and light conditions and were fed laboratory chow and water ad libitum. Mice ($n = 5$/group) housed in the same cage were randomly assigned to receive an intravenous injection of 200 µl of an arthritic dose of either *S. aureus* Newman strain ($2.8 \times 10^6$ CFU/mouse) or $\Delta nuc1$ mutant ($2.8 \times 10^6$ CFU/mouse). Mice were monitored for weight loss and clinical signs of arthritis from day 0 to day 7 in a manner blinded to the bacterial strains. Morphine (Abcur AB, 10 mg/kg) was administered subcutaneously (s.c.) daily to all mice starting three days after infection to alleviate pain associated with septic arthritis. On day 7, mice were sacrificed to collect samples, including blood, kidneys, and joints. Kidneys were collected aseptically and scored on a scale of 0 (no abscess), 1 (mild abscess), 2 (moderate abscess) to 3 (severe abscess). The kidneys were then minced and diluted with sterile PBS. Dilutions were plated on horse blood agar plates, incubated at 37 °C for 24 h, and the colonies obtained were counted using a colony counter (Stuart Scientific, Made in the UK).

### Clinical evaluation of arthritis
Observers (M.D. and T.J.) blinded to the treatment groups visually inspected all 4 limbs of each mouse. Arthritis was defined as erythema and/or swelling of the joints. A clinical scoring system ranging from 0 to 3 was used for each paw (0, no inflammation; 1, mild visible swelling and/or erythema; 2, moderate swelling and/or erythema; and 3, marked swelling and/or erythema).

### Microcomputed Tomography (µCT)
On 7 day post-infection, the mice were sacrificed and all paws were scanned by SkyScan 1176 µCT (Bruker, Antwerp, Belgium). The scanning was conducted at 55 kV/ 455 µA, with a 0.2-mm aluminum filter. The exposure time was 47 ms. The X-ray projections were obtained at 0.7° intervals with a scanning angular rotation of 180°. The NRecon software (version 1.6.9.8; Bruker) was used to reconstruct 3D images which were further evaluated by using CT-Analyzer (version 2.7.0; Bruker). Each joint was evaluated by two researchers (M.D. and T.J.), in a blinded manner, using a scoring system from 0 to 3 (0: healthy joint; 1: mild bone destruction; 2: moderate bone destruction; and 3: marked bone destruction) as previously described[17].

### NucA expression and purification
NucA is an extracellular enzyme, which is secreted as mature Nuc1. The *nucA* gene was cloned into the vector pET28a with C-terminal His-tag and this construct was transformed into *E. coli* BL21(DE3). The transformant carrying pET28a-NucA-6xHis was grown in LB at 37 °C supplemented with 50 µg/ml ampicillin. When OD600 reached 0.6–0.8, the bacteria were induced with 1 mM IPTG at 18 °C overnight for overexpressing NucA. The overnight culture was collected, resuspended in buffer A (50 mM Tris-HCl pH 8.0, 300 mM NaCl), and lysed by an ultrasonic sonicator with a pulse every 4 s for 4 min. The lysate was centrifuged at $14,000 \times g$ for 1 h. The supernatant of lysate was collected and then loaded onto an Ni-NTA column. Fractions containing NucA were collected with Buffer B (20 mM

Tris-HCl pH 8.0, 200 mM NaCl, 250 mM imidazole) and analyzed by SDS-PAGE. NucA enzyme was dialyzed with PBS, concentrated, flash-frozen in liquid nitrogen, and stored at -80 °C until use.

### Genomic bacterial DNA (gDNA) degradation assay with NucA
*S. aureus* JE2$\Delta lgt$ was cultured in TSB medium overnight. The bacterial pellet was collected and resuspended in TE buffer supplemented with lysostaphin and RNase at 37 °C for 30 min. Bacterial gDNA was then purified via phenol-chloroform-isoamyl alcohol, precipitated with iso-propanol, washed with ethanol, and then dissolved in $H_2O$[69]. gDNA was incubated with varying concentrations of recombinant NucA for 1 h at 37 °C. Samples were then added to RAW 264.7 cells for stimulation and loaded on the agarose gel for visualization.

### Preparation of bacteria and bacterial supernatant
BM medium was used for inoculating *S. aureus* at 37 °C with shaking from fresh BM agar plates. Cultures were harvested after 16 h by centrifuging and washed with PBS. To get bacterial dosage (MOI, multiplicity of infection), bacteria were calculated to OD/CFU. Bacterial supernatants were collected and filtered with a 0.2 µm pyrogen-free round column. The supernatants were kept on ice until use and adjusted to equal concentrations according to bacterial number. The supernatant was tested for its ability to degrade gDNA and its activity on neutrophils.

### Neutrophil isolation
Venous blood was freshly collected by EDTA-tubes (Sarstedt, Germany) from several healthy individuals. 6 mL blood was layered on 6 mL of Lymphocyte poly-cell separation medium (Cedarlane, Burlington, Canada). Centrifugation was done without pause, at $500 \times g$ for 40 min at room temperature. The PMN layer was collected and washed twice with 12 mL PBS, and centrifuged at $400 \times g$ for 10 min at room temperature with settings of acceleration 5 and deceleration 4. Cells were resuspended in RPMI medium without phenol red (Sigma-Aldrich, Darmstadt, Germany). Cell counts were obtained by the Trypan Blue exclusion method, utilizing a Neubauer counting chamber.

### Cell culture and immune stimulation assay
The murine macrophage cell line RAW 264.7 was cultured in Dulbecco's modified Eagle's medium (gibco) supplemented with 10% fetal bovine serum (FBS) and 1% penicillin-streptomycin at 37 °C with 5% $CO_2$. The human osteoblast-like cell line SAOS-2 was grown in McCoy's 5 A Medium (Sigma) supplemented with 15% FBS, 1% glucose, and 1% penicillin-streptomycin. Prior to stimulation, RAW 264.7 and SAOS-2 cells were seeded in 96-well plates and incubated overnight until reaching confluency, while neutrophils were directly seeded into the plates before adding all stimulants. Immune stimulation was performed for 18 h at 37 °C and 5% $CO_2$, except for neutrophils which were stimulated for 5 h. The cellular supernatants were then collected and stored at $-20$ °C until determining the cytokines production.

### Detection of cytokines by ELISA
Cytokines collected from the cellular supernatants of different cell lines were measured with the uncoated ELISA kit (Invitrogen) according to instructions. The plasma levels of IL-6, TNF-α, keratinocyte chemoattractant (KC), and S100A8/A9 in blood collected from NMRI mice intravenously infected with Newman WT (NWT) and $\Delta nuc1$ were quantified using DuoSet ELISA kits (R&D Systems Europe) according to the manufacturer's instructions.

### Neutrophil bacterial killing assay
JE2 and JE2$\Delta nuc1$ were grown overnight. Bacteria were collected, regrown to the log phase, washed with PBS, and opsonized with 10% human pooled serum in RPMI for 1 h at 37 °C. Neutrophils were seeded in 24-well plates, incubated with bacteria (100%) at an MOI = 0.1. After 10 min, 20 min, 30 min, and 1 h, the neutrophils were lysed with ice-cold ddH$_2$O and centrifuged for 15 min at 4 °C. The lysates were plated on agar plates with serial

dilutions and CFUs were counted the next day. The *S. aureus* survival (killing) was calculated by comparing the counted CFU to the original added CFU and normalized to JE2.

## Phagocytosis assay

Bacteria were grown overnight, collected, regrown until the log phase, and washed with PBS. Bacteria were opsonized with 10% human pooled serum in RPMI without phenol red for 1 h at 37 °C and then were labeled with Alexa Flour 633 conjugate (Invitrogen, W21404) for 20 min at 37 °C. The unlabeled fluorochrome was washed twice by PBS. Neutrophils were seeded into 24-well plates and incubated with bacteria at an MOI = 2. After 1 h incubation, the neutrophils were fixed with 3.7% formaldehyde for 20 min on ice. The fluorescence intensity of fixed neutrophils was determined with a BD FACSCalibur (BD Biosciences). The phagocytotic index was calculated as number of fluorescent-positive neutrophils multiplied by the fluorescence mean and normalized to JE2 parent strain to minimize the individual error. This index shows how many bacteria were phagocytosed per cell.

## Internalization assay

RAW 264.7 cells were seeded in 24-well plates with 500 µl of culture medium until reaching confluency. Cells were washed with PBS, and the pre-warmed culture medium without antibiotics was added to each well. Bacteria were grown to the log phase before infection. The cells were then incubated with bacteria for 1.5 h to yield an MOI of 20:1. After incubation, gentamycin, and lysostaphin were added to kill the extracellular bacteria for 1 h. The cells were lysed with 0.1% Triton X-100 supplemented with 0.05% Trypsin and lysates were plated to determine internalized bacteria[70]. Internalization was calculated as CFU of internalized bacteria/host cell-seeded and normalized to JE2.

## Sytox green assay

Isolated neutrophils were prepared to be $2 \times 10^5$ cells/mL, and then 1 µM Sytox Green (Thermo Fisher, Waltham, USA) was added. 1% Triton X-100 solution was used for extracellular DNA normalization. Cells were stimulated for 5 h with MOI = 1:2 live bacteria and 2% overnight bacterial supernatant. Fluorescent intensity was measured every 30 min at Ex 485 nm/Em 520 nm, and the cells were incubated at 37 °C with 5% $CO_2$ by the microplate reader (FluoStar Omega, BMG Labtech, Ortenberg, Germany). Microscopic images were taken with EVOS Fl (Thermo Fisher) fluorescence microscope at 3 h of incubation.

## Live-dead staining

Isolated neutrophils were incubated with an MOI of 1:2 live bacteria and 2% overnight bacterial supernatant. To visualize live cells, neutrophils were stained with 2 µM Calcein AM and Ethidium Bromide for 10 min at 37 °C in RPMI medium without phenol red. Images were taken with the EVOS Fl (Thermo Fisher) fluorescence microscope.

## Immunofluorescence

Neutrophils were diluted to $3 \times 10^5$ cells/mL and seeded onto self-prepared poly-L-lysine coated chamber slides. The cells were incubated with live bacteria at an MOI of 2 and 2% overnight bacterial supernatant at 37 °C in a 5% $CO_2$ for 1 h. Following incubation, the cells were fixed with 4% formaldehyde and permeabilized using 0.5% Triton X-100. After blocking with 5% bovine serum albumin (BSA) in PBS, the cells were incubated overnight with myeloperoxidase (1:200 in PBS, sc-52707, Santa Cruz Biotechnology, Heidelberg, Germany) and citH3 (1:1000 in PBS, ab5103, Abcam). After washing with PBS, the staining was continued with an Alexa Fluor-488 conjugated secondary antibody (1:1000 in PBS, A10667, Invitrogen, Carlsbad, CA, USA) and Hoechst 33342 (2 µg/mL) for 2 h. The chambers were then removed, and the slides were mounted using Fluoromount G mounting medium (Thermo Fisher) and covered with coverslip. Microscopy was conducted using an EVOS Fl fluorescence microscope (Thermo Fisher).

## Ethical statement

The Ethics Committee of Animal Research of Gothenburg approved all experiments conducted on mice. The mouse experiments were performed in accordance with the Swedish Board of Agriculture's regulations and recommendations on animal experiments. We have complied with all relevant ethical regulations for animal use. Blood was collected from healthy adult volunteers and written informed consent was given. The institutional review board of the University of Tübingen approved the study and all adult subjects provided informed consent. This study was done in accordance with the ethics committee of the medical faculty of the University of Tübingen that approved the study (Approval number 015/2014 BO2). All ethical regulations relevant to human research participants were followed.

## Statistics and reproducibility

All the data were analyzed using GraphPad Prism (version 6.0; GraphPad Software). The data are presented in mean ± standard error of the mean (SEM). Statistical significance: ns (not significant) $p > 0.05$; *$p < 0.05$; **$p < 0.01$; ***$p < 0.001$; ****$p < 0.0001$. Details of statistical analyses for each experiment are provided in "Materials and Methods".

## Reporting summary

Further information on research design is available in the Nature Portfolio Reporting Summary linked to this article.

## Data availability

Primary source data are provided in Supplementary Data 1. Additional requests for the data and materials in this study are available from the corresponding author upon reasonable request.

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

## Acknowledgements

This work was supported by funding from the Deutsche Forschungsgemeinschaft the Germany's Excellence Strategy—EXC 2124—390838134 "Controlling Microbes to Fight Infections" to F.G., grants from the Swedish state under the agreement between the Swedish Government and the county councils, the ALF-agreement grant number ALFGBG-823941 to T.J., N.L. was supported by the Chinese Scholarship Council. We are grateful to Stefanie Krajewski, Clinical Research Laboratory, Department of Thoracic, Cardiac and Vascular Surgery, University Hospital Tübingen, 72076 Tübingen, Germany, for providing us with SAOS-2 cells. S.E. received funding from the German Research Council (EH471/5-1). The funders had no role in design, analysis, and reporting of the study. A.W. received funding from the DFG Collaborative Research Center (CRC) 156 "The skin as a sensor and effector organ orchestrating local and systemic immune responses" (project B05) and DFG Priority Programs SPP 2225 "EXIT Strategies of Intracellular Pathogens" (projects 446404928 and We-4195/25-1). We acknowledge support by Open Access Publishing Fund of University of Tübingen.

## Author contributions

F.G., N.L. designed the study and the experiments; T.J., M.V.D. designed and carried out the septic arthritis model; F.S., S.E., A.N., and N.L. carried out neutrophil experiments; N.L., N.H. constructed deletion mutants; N.L., A.V.A., and A.W. carried out cell lines experiments; F.G., N.L. wrote the manuscript. All authors read and approved the final manuscript.

## Funding

## Competing interests

The authors declare no competing interests.
