## [Transparent Peer Review file · Communications Biology]

Staphylococcus aureus thermonuclease NucA is a key virulence factor in septic arthritis

Corresponding Author: Professor Friedrich Götz

Version 0:

Reviewer comments:

Reviewer #1

(Remarks to the Author)

Thank you for the opportunity to review this paper titled "Staphylococcus aureus thermonuclease NucA is a key virulence factor in septic arthritis". The manuscript reports that in the mouse model for hematogenous septic arthritis 23 the Δ nuc1 mutant was much less pathogenic and the severity of clinical septic arthritis 24 was markedly reduced, including decreased weight loss, lower kidney bacterial loads 25 and much less IL-6 production. Additionally, the manuscript demonstrates that NucA induced higher IL-6 production in SAOS-2 and higher TNF-28 α and IL-10 production in neutrophils and shielded *S. aureus* from phagocyte 29 engulfment and killing. This is an important area of investigation as septic arthritis is an aggressive joint disease causing significant morbidity and mortality. Below I have suggested some modifications and additional experiments to strengthen the manuscript. I wish you the best with this submission.

1. The manuscript would benefit from further discussion of the rationale for focusing on TNF in later figures given that Figure 2D demonstrates minimal differences in TNF production between NWT and the nuc1 deletion mutant.

2. In the discussion of Figure 3 the manuscript states that both ODN2006 and ODN2137 were employed in panel 3A (Lines 134-136). However, the data for ODN2137 was not included and is an important negative control.

3. The manuscript would be strengthened from documenting experimentally or referencing previous characterization of the strains employed (JE2, JE2 Δ nuc1, JE2 Δ Igt, JE2 Δ nuc1 Δ Igt) in terms of nuc1 released by bacteria and capacity to degrade DNA. Additionally, it is necessary to characterize the presence of extracellular DNA available for nuc1 digestion. Currently, the data in figure 4 could be alternatively explained by differences in nuc1 release, nuc1 activity, or differences in available extracellular DNA during infection of RAW 264.7, SAOS-2, and Neutrophils. In particular, for infection of neutrophils are NETs formed for this MOI at this time point?

4. The manuscript examines bacterial killing in various cell types in Figure 5. In particular the data indicate a significant reduction in bacterial survival in neutrophils for the JE2 Δ nuc1, JE2 Δ Igt, JE2 Δ nuc1 Δ Igt strains compared to JE2. The manuscript indicates this is due to differences in internalization. However, it does examine internalization in the context of neutrophils only RAW 264.7 and SAOS-2.

5. In figure 5B-C presents data indicating reduced recovery of intracellular viable CFU for the JE2 Δ nuc1, JE2 Δ Igt, JE2 Δ nuc1 Δ Igt strains compared to JE2 suggesting a reduction in internalization. Additional immunofluorescence experiments should be conducted to confirm that this is in fact due to internalization. With the current experimental the mutants could display the same degree of internalization as the parent JE2 but simply be more susceptible to intracellular killing.

6. The manuscript states in the discussion that NucA shields *S. aureus* from phagocyte engulfment. Additional experiments should be included to fully demonstrate that there is reduced internalization with the JE2 Δ nuc1 strain compared to the parent JE2. It would further support this statement to complement the JE2 Δ nuc1 infection with nuclease to demonstrate that this can restore internalization and survival levels to that observed for the JE2 parent.

7. The manuscript would benefit from a schematic documenting the authors' current results in the context of septic arthritis.

Reviewer #2

(Remarks to the Author)

This is a very interesting manuscript detailing clearly that NucA is a significant factor in *S. aureus* septic arthritis. This was documented using well described animal models of infection. It was interesting to note, however, that while NucA is also important in abscess formation in the kidney, it may not be important in metastasis through the vasculature or liver as the same bacterial burden was noted in the kidney.

It was also very clear that NucA functions to degrade staphylococcal DNA and in addition eDNA/RNA from the host. However, there were other aspects of the manuscript that seemed disjointed. First, it was unclear why the investigators utilized various cell lines in this manuscript—it was not clear to the reader why RAW 264.7 and SAOS-2 cells were used and how one can interpret the different cytokine responses that were noted.

Second, although it was discussed that one potential reason NucA is so important in septic arthritis is that it induces a significant IL-6 response, the investigators did not use an IL-6 knockout mouse to address this hypothesis. Unless there are other mitigating factors that are not understood, this seems to be an important point.

Minor comments:

the authors might want to replace massive with significant or another word throughout the manuscript

I am not familiar with the term diffuse areal and was unable to find this term in the literature (or at least associated with biology)?

Unclear that the discussion from lines 272-280 are needed. In my opinion, no need to justify why another strain was not used.

Neutrophils instead of neutrophiles in line 31

Reviewer #3

(Remarks to the Author)

Summary. In this manuscript authors demonstrate a role for the extracellular nuclease, NucA in the pathogenesis of *S. aureus* during septic arthritis. This was found to be associated with the ability of NucA to prevent the induction of an appropriate, inflammatory immune response to *S. aureus*. The authors demonstrate a very interesting phenotype for bone erosion caused when NucA is expressed in vivo, that may be related to the levels of inflammation induced in response to *S. aureus* DNA, providing exciting avenues for further research.

Key Strengths. To the best of my knowledge, this is the first report for the role of nucA in septic arthritis and control of local inflammation caused therein. The methods used in the manuscript are well established and robust.

Key Weaknesses. The writing is very confusing. Rationale for certain experiments (and conclusions made thereof) is unclear.

Major Comments.

1. The writing of this manuscript and the grammatical style is very difficult to follow, and tense uses make interpretation of the data tenuous. There are also a number of abbreviations used throughout the manuscript that are never introduced. What are NMRI mice? What are SAOS-2 cells and why are they relevant here? Additionally, an introduction of terms such as 'clinical arthritis frequency' and 'clinical arthritis score' would aid the interpretation of results.
2. The authors switch back and forth between strains. Primary mouse experiments are done in Newman, the next few in vitro experiments switch to SA113 and then to JE2 (USA300). The justification for this is completely lost on the reader and should be explained somewhere other than the discussion.
3. The authors pick 4 cytokines to measure in Figure 2. There is no clear explanation of why these were chosen. These cytokines change in Figure 4. There is once again no justification as to why this panel changed.
4. Line 122- 123 starts this section by explaining the role of TLR-9. TLR9 however never makes another appearance in the manuscript, other than in the discussion where it also feels completely disconnected.
5. It is difficult to reconcile Figure 1d with 1e. While there is a large decrease in the abscess score for the nuclease mutant this translates to a less than 1 log drop of cfu. Images of kidneys to show abscesses would assist the interpretation of this data. Authors should consider additionally checking the synovial fluid for CFU.
6. Line 57-58 mentions biofilms but has no relevance to the rest of the manuscript. NucA has been demonstrated to break down NETs that are induced by *S. aureus* biofilms. These phenotypes/studies should be addressed especially since septic arthritis is associated with biofilm formation.
7. Studies demonstrate that *S. aureus* can kill primary human neutrophils as quickly as 30 minutes following incubation. It is unclear how many neutrophils are alive at 2 hours or whether neutrophils are still alive at 5 hours post incubation. DNA can be released passively from neutrophils at this point. An earlier time point might provide clearer differences between strains, for NET release. The source of the Sytox labelled DNA is also unclear (PMN/bacterial). Authors should consider measuring MPO, neutrophil elastase and citrullinated histones as a measure of NETosis. Labelling of neutrophils with Cell Tracker would facilitate interpretation.

8. Previous studies demonstrate multiple phenotypes associated with deoxyadenosine generated following NET release, yet this not clearly addressed here. Does the release of neutrophil extracellular traps cause macrophage cytotoxicity as previously described? Considering previous studies were also performed in an intravenous model of sepsis, it would be important to exemplify the differences (and similarities) observed with some of these phenotypes.

9. Since a number of surface proteins have been demonstrated to contribute to virulence in septic models, it is especially relevant that isogenic mutants used in this manuscript be complemented in more than one assay.

10. Is Figure 2a an n=1? Differences in Figure 2c are very difficult to see. Consider a zoom in box to the ROI. Since the novelty here, compared to previous studies, is the bone erosion phenotype, this should be clearly depicted. Images of complemented strain infection would solidify this further, especially since Newman is an osteomyelitis isolate.

11. MOI is 1:2 (presumably) but stated as MOI:2. It is also unclear whether neutrophils are 1 or 2. The methods state an MOI of 0.1 for neutrophil assays. The choice and reasons for change should be stated.

12. The rationale for Figure 5 is unclear. What is the difference between a killing assay and an uptake assay and why was uptake not performed in neutrophils or killing in macrophages/SAOS cells? Uptake cannot be concluded from this data unless it is clear that RAW and SAOS cells do not kill bacteria.

13. For Figure 5, it is unclear how these CFUs are calculated. The methods mention plating (assuming invasion assay is the uptake assay), but the axis says CFU/cell. An appropriate measurement would be percent of time 0.

Line 173 mentions a 2-hour incubation, the legend says 1.5hrs

14. Line 239- 242- Conclusions made here are far reaching and should be re-phrased. The authors conclude that NucA protects *S. aureus* from 1) engulfment- this is not shown in this manuscript and 2) killing- at the time point shown for neutrophils, only 20% of WT USA300 survives. That's not much of an advantage and could presumably reduce if left longer. Additionally, NucA degrading NETs is published, it is not a novel suggestion from data in this manuscript (Thammavongsa et al, 2013). This should be re-phrased.

Minor Comments

1. Lines 192- 202 are very confusing and very difficult to interpret.

2. There are numerous typos in the manuscript such as line 31, line 81, line 89 (either/or not both/or), line 150, supplement line 9, line 300, line 302, line 305

3. Is Figure 2a an n=1? Differences in Figure 2c are very difficult to see. Consider a zoom in box to the ROI.

Version 1:

Reviewer comments:

Reviewer #1

(Remarks to the Author)

Thank you for the opportunity to review this paper titled "Staphylococcus aureus thermonuclease NucA is a key virulence factor in septic arthritis". The manuscript reports that in the mouse model for hematogenous septic arthritis 23 the Δ nuc1 mutant was much less pathogenic and the severity of clinical septic arthritis 24 was markedly reduced, including decreased weight loss, lower kidney bacterial loads 25 and much less IL-6 production. Additionally, the manuscript demonstrates that NucA induced higher IL-6 production in SAOS-2 and higher TNF-28 α and IL-10 production in neutrophils and shielded *S. aureus* from phagocyte 29 engulfment and killing. This is an important area of investigation as septic arthritis is an aggressive joint disease causing significant morbidity and mortality. The authors were highly responsive to my previous suggestions and have incorporated sufficient additional data and information to alleviate all of my previous concerns.

Reviewer #2

(Remarks to the Author)

No further comments, the investigators addressed my comments appropriately. Outstanding work.

Reviewer #3

(Remarks to the Author)

This manuscript demonstrates the role of the nuclease nuc1, as a virulence factor in a murine model of septic arthritis caused by *S. aureus*. I feel that this manuscript adds to our knowledge of the pathogenesis of *S. aureus* and will be valuable to the field. However, the writing would greatly benefit from an in-depth review of grammatical errors which includes numerous spelling errors and more importantly, sentence framing. These errors take away from the value of the data presented in this story.

For example, line 352- 354- Although the in vivo experiments revealed little differences in TNF α induction in response to the parental and the mutant, we decided to verify this observation also in cell lines.

Suggestion- While in vivo experiments showed minimal differences in TNF- α induction following infection with either WT or

the isogenic nuclease mutant, we further evaluated these phenotypes in vitro and found that...
Another example: Line 361- 363- Sentence abruptly ends.

Line 27-28- Typo. 'In vitro, *S. aureus* genomic DNA 27 induced a robust TNF- α response in macrophage-like cells was? abrogated when the DNA 28 was degraded by NucA'.

Line 29- Respectively spelled wrong

Line 97- M&M should not be used in a manuscript in place of full words.

Figure 1 legend should be re-phrased. 'More clinical arthritis' should be replaced with increased severity or something similar.

I still think Figure 1d would benefit from images of kidney abscesses and that these should always be shown but if time is an issue, that's fine.

Line 124. Consider removing the word massive

Line 126- 127. The reasoning for this selection of cytokines needs to be introduced. One could argue for a number of other pro-inflammatory cytokines such as IL-1b, GCSF and GMCSF to be measured in the absence of this reasoning. Mentioning this in the discussion does not aid the interpretation of the results.

Line 148- Virulence

Line 158- The reader is not aware of what JE2 is, furthermore the correct annotation for the first time this strain is introduced, is USA300 JE2. (Mentioned later in line 172 and then switches back).

Line 172- 173. This is an overstated sentence. There are many studies where USA300 and Newman do not align on virulence properties, and the results depend on the infection context. This should be stated here.

Line 175- 177. It seems like the introduction of the two types of nucleases is relevant at the beginning of the paper where the authors first mention nuc1 as the virulence factor of interest.

Line 199- Typo. Cytokine not cytokines

Line 207- This is the first time these cytokines are introduced. The equivalent of KC would be IL-8. IL-10 is not classically pro-inflammatory. The selection of IL-1Ra is unclear to me. The discussion provides a very detailed explanation of TNF- α and IL-6 but still doesn't tell me about these other cytokines. Additionally, introducing the context at the end of the paper diminishes its value for data interpretation. The revised text in the manuscript just says TNF. That's not a cytokine.

Line 212-218. These are very strong assumptions made about the relative virulence differences between Newman and USA300 that are not supported by the data. Cytokine readouts (especially choosing 3) can be similar for many reasons and can elicit vastly different local and systemic responses during infection. Considering these macrophage and bone cells are not primary and that these phenotypes are not genetically complemented, the language of these statement is too strong and needs to be dialed back.

Line 231- 233. What is this assumption based on? Literature? Reference required.

Line 244- The complementation was made on an episome?

Figure 5 is very confusing. The legend mentions that RFU are plotted in A, but of what, is completely lost. The red anti-citrullinated histone stain is not easy to see, the channels should be separated in order for the red channel to be highlighted. The novelty of these findings is also unclear <https://pmc.ncbi.nlm.nih.gov/articles/PMC2982853/>. Explaining the reasoning for these experiments in the context of this story, at the beginning of this section would assist the manuscript.

Line 276. Representative images should be demonstrated, even when not exemplary.

Line 285- The word massive is not scientific or specific and should not be used to describe data.

Line 310- 318. If TLR-9 is the only thing that initiates a host response to *S. aureus* please provide references of this statement. This entire paragraph has no references.

Line 330- 336. Are there any references for this paragraph?

Line 527- Typo. Neutrophil not neutrophils

Supplement lines 28-29. Grammar

Point-by-point response to reviewers

Reviewer #1 (Remarks to the Author):

Thank you for the opportunity to review this paper titled “Staphylococcus aureus thermonuclease NucA is a key virulence factor in septic arthritis”. The manuscript reports that in the mouse model for hematogenous septic arthritis 23 the $\Delta nuc1$ mutant was much less pathogenic and the severity of clinical septic arthritis 24 was markedly reduced, including decreased weight loss, lower kidney bacterial loads 25 and much less IL-6 production. Additionally, the manuscript demonstrates that NucA induced higher IL-6 production in SAOS-2 and higher TNF-28 α and IL-10 production in neutrophils and shielded *S. aureus* from phagocyte 29 engulfment and killing. This is an important area of investigation as septic arthritis is an aggressive joint disease causing significant morbidity and mortality. Below I have suggested some modifications and additional experiments to strengthen the manuscript. I wish you the best with this submission.

1. The manuscript would benefit from further discussion of the rationale for focusing on TNF in later figures given that Figure 2D demonstrates minimal differences in TNF production between NWT and the *nuc1* deletion mutant.

Thank you for pointing this out. Although the *in vivo* experiments revealed little differences in TNF- α induction in response to the parental and the mutant, we wanted to verify this observation also in cell lines because this cytokine is strongly released by macrophages in response to conserved bacterial components such as peptidoglycan, lipoprotein, protein A and DNA.

We have included the text in the discussion about the TNF cytokines as follows:
Line (352-358): Although the *in vivo* experiments revealed little differences in TNF- α induction in response to the parental and the mutant, we decided to verify this observation also in cell lines. Consistent with the *in vivo* data, we observed no notable differences in TNF- α production by macrophage-like and osteoblast-like cells when incubated with JE2 and JE2 $\Delta nuc1$ (Figure 4a, b and Supplementary Figure 3). However, in neutrophils more TNF- α production was induced by the parental strain than the *nuc1* mutant.

2. In the discussion of Figure 3 the manuscript states that both ODN2006 and ODN2137 were employed in panel 3A (Lines 134-136). However, the data for ODN2137 was not included and is an important negative control.

Thanks for pointing this out. In this study, standard CpG oligonucleotides ODN2006 and the non-CpG oligonucleotides ODN2137 were employed as positive and negative controls, respectively, to illustrate bacterial DNA-induced release of TNF- α in RAW 264.7 cells. But we forgot to include the negative control ODN2137 in Fig. 3. We apologize for overlooking this. We now include ODN2137 in the revised version of Fig. 3a.

3. The manuscript would be strengthened from documenting experimentally or referencing previous characterization of the strains employed (JE2, JE2 $\Delta nuc1$,

JE2 Δ lgt, JE2 Δ nuc1 Δ lgt) in terms of nuc1 released by bacteria and capacity to degrade DNA. Additionally, it is necessary to characterize the presence of extracellular DNA available for nuc1 digestion. Currently, the data in figure 4 could be alternatively explained by differences in nuc1 release, nuc1 activity, or differences in available extracellular DNA during infection of RAW 264.7, SAOS-2, and Neutrophils. In particular, for infection of neutrophils are NETs formed for this MOI at this time point?

We have updated the text as follows to illustrate the difference in NucA activity across the used strains.

Line (175-183): Of the two nuclease genes (*nuc1* and *nuc2*) in *S. aureus*, it is the *nuc1*-encoded NucA that is secreted into the supernatant and is able to degrade extracellular DNA and RNA (eDNA and eRNA, respectively)²¹. This was demonstrated when we incubated the supernatant of JE2 and its mutants JE2 Δ nuc1, JE2 Δ lgt, and JE2 Δ nuc1 Δ lgt with *S. aureus* gDNA for 1 h to evaluate the nuclease activity. In all mutants in which *nuc1* was deleted, the gDNA remained intact, whereas in JE2 and the JE2 Δ lgt mutant the gDNA was completely degraded, indicating that NucA is responsible for gDNA degradation (Supplementary Figure 1a).

The question as to differences in available extracellular DNA during infection of RAW 264.7, SAOS-2, and neutrophils is difficult to understand because the reviewer refers to Figure 4, where we show cytokine quantification in response to the various bacterial strains. We quantify NET formation and degradation in neutrophils only (this is figure 6) by fluorescently staining the DNA in NETs, which is shown in blue in the picture. The supernatants from JE2 Δ nuc1 and JE2 Δ nuc1 Δ lgt triggered more eDNA as compared to JE2 and JE2 Δ lgt, indicating that NucA is responsible for eDNA degradation. In this experiment, we used a MOI of 2, which induces very little NETs formation after 1 h (Fig. 6c).

4. The manuscript examines bacterial killing in various cell types in Figure 5. In particular the data indicate a significant reduction in bacterial survival in neutrophils for the JE2 Δ nuc1, JE2 Δ lgt, JE2 Δ nuc1 Δ lgt strains compared to JE2. The manuscript indicates this is due to differences in internalization. However, it does examine internalization in the context of neutrophils only RAW 264.7 and SAOS-2.

We agree that investigating the internalization in neutrophils could give us more valuable information. We have now performed a time-course survival assay in neutrophils, which revealed that JE2 Δ nuc1 exhibited decreased survival as compared to the wild type strain already 10 minutes after exposure to neutrophils (revised Fig. 5a), suggesting that its decreased internalization 1h after exposure to neutrophils likely results from increased killing. Similar results were obtained in RAW 264.7 cells.

5. In figure 5B-C presents data indicating reduced recovery of intracellular viable CFU for the JE2 Δ nuc1, JE2 Δ lgt, JE2 Δ nuc1 Δ lgt strains compared to JE2 suggesting a reduction in internalization. Additional immunofluorescence experiments should be conducted to confirm that this is in fact due to internalization. With the current experimental the mutants could display the same degree of internalization as the parent JE2 but simply be more susceptible to intracellular killing.

See the reply to point 4.

6. The manuscript states in the discussion that NucA shields *S. aureus* from phagocyte engulfment. Additional experiments should be included to fully demonstrate that there is reduced internalization with the JE2 Δ nuc1 strain compared to the parent JE2. It would further support this statement to complement the JE2 Δ nuc1 infection with nuclease to demonstrate that this can restore internalization and survival levels to that observed for the JE2 parent.

We have now added experiments with JE2, JE Δ nuc1 and complemented strain JE2 Δ nuc1(pRB473-nuc1) to confirm the role of NucA. The JE2 Δ nuc1(pRB473-nuc1) formed a dark blue hole in the DNase Test agar indicating that the nuclease ability was restored (Supplementary Figure 1d). Fig. 5 shows that the phagocytosis and bacterial survival were restored in the complemented strains when incubated with neutrophils. Additionally, the complemented Δ nuc1 mutant showed a similar internalization pattern as the parent strain in RAW 264.7.

7. The manuscript would benefit from a schematic documenting the authors' current results in the context of septic arthritis.

It is a good suggestion. We have included a schematic representation to explain our results about septic arthritis.

Reviewer #2 (Remarks to the Author):

This is a very interesting manuscript detailing clearly that NucA is a significant factor in *S. aureus* septic arthritis. This was documented using well described animal models of infection. It was interesting to note, however, that while NucA is also important in abscess formation in the kidney, it may not be important in metastasis through the vasculature or liver as the same bacterial burden was noted in the kidney.

It was also very clear that NucA functions to degrade staphylococcal DNA and in addition eDNA/RNA from the host. However, there were other aspects of the manuscript that seemed disjointed. First, it was unclear why the investigators utilized various cell lines in this manuscript--it was not clear to the reader why RAW 264.7 and SAOS-2 cells were used and how one can interpret the different cytokine responses that were noted.

Thanks for your thoughtful feedback. We agree that we did not clearly explain the rationale behind the use of the various cell lines. In our mouse model, NWT was observed to cause bone erosion, whereas Δ nuc1 did not. Therefore, we assume the innate immune cells, such as macrophages and neutrophils, and other cell lines, such as osteoblasts and osteoclasts, may play a role in the observed phenotype. Accordingly, macrophages RAW 264.7 and osteoblasts SAOS-2 were employed.

We have now included the explanation about the cell lines used in this manuscript as follows:

Line (200-203): *S. aureus* activates different host cells, including macrophages, neutrophils and osteoblasts²²⁻²⁴. We further investigated whether the immune stimulation of JE2 and its mutants differed in RAW 264.7, neutrophils and osteoblast-like SAOS-2 cells.

Line (333-337): *S. aureus* stimulates the immune response in various cells, including macrophages and neutrophils, as well as non-immune cells, such as osteoblasts cells. In addition, osteoblast and osteoclast cells are responsible for bone remodeling and construction. Accordingly, macrophages RAW 264.7, osteoblast cells SAOS-2 and neutrophils were employed in this study.

The differences in IL-6 production between RAW 264.7 and SAOS-2, with the former producing higher amounts, clearly reflect the important role of macrophages in the response to bacteria as compared to osteoblasts, which is supported by other studies.

Second, although it was discussed that one potential reason NucA is so important in septic arthritis is that it induces a significant IL-6 response, the investigators did not use an IL-6 knockout mouse to address this hypothesis. Unless there are other mitigating factors that are not understood, this seems to be an important point.

In the $\Delta nuc1$ -infected mice, largely decreased IL-6 and S100A8/A9 were observed. We totally agree that IL-6 plays a role in this process. Other groups have shown that a lack of IL-6 would lead to decreased arthritis scores and bone erosion. High concentration of IL-6 is observed in patients with septic arthritis. However, analyzing the IL-6 knockout mouse in the limited revision time we have available is not feasible. We will do it in the future.

Minor comments:

the authors might want to replace massive with significant or another word throughout the manuscript

We have replaced massive with significant, markable and notable.

I am not familiar with the term diffuse areal and was unable to find this term in the literature (or at least associated with biology)?

We have re-phased the whole paragraph and Figure 6 to make it clearer.

Unclear that the discussion from lines 272-280 are needed. In my opinion, no need to justify why another strain was not used.

We have deleted it and provided a brief mention (172-175) in Results.

Neutrophils instead of neutrophiles in line 31.

We have corrected it.

Reviewer #3 (Remarks to the Author):

Summary. In this manuscript authors demonstrate a role for the extracellular nuclease, NucA in the pathogenesis of *S. aureus* during septic arthritis. This was found to be associated with the ability of NucA to prevent the induction of an appropriate, inflammatory immune response to *S. aureus*. The authors demonstrate a very interesting phenotype for bone erosion caused when NucA is expressed in vivo, that may be related to the levels of inflammation induced in response to *S. aureus* DNA, providing exciting avenues for further research.

Key Strengths. To the best of my knowledge, this is the first report for the role of nucA in septic arthritis and control of local inflammation caused therein. The methods used in the manuscript are well established and robust.

Key Weaknesses. The writing is very confusing. Rationale for certain experiments (and conclusions made thereof) is unclear.

Major Comments.

1. The writing of this manuscript and the grammatical style is very difficult to follow, and tense uses make interpretation of the data tenuous. There are also a number of abbreviations used throughout the manuscript that are never introduced. What are interna mice? What are SAOS-2 cells and why are they relevant here? Additionally, an introduction of terms such as 'clinical arthritis frequency' and 'clinical arthritis score' would aid the interpretation of results.

We appreciate the constructive feedback and apologize for the grammatical errors. We have carefully revised the manuscript, especially the conclusion part, and added the explanation to abbreviations and relevance in cell lines and mouse model.

Naval Medical Research Institute (NMRI) mice are an outbred mouse strain with larger genetic variations, resembling the clinical situation. They were employed in our study.

SAOS-2 cells are osteoblast cells. Osteoblasts join in bone remodeling and immune response. Previous studies have investigated whether *S. aureus* could invade osteoblasts and induce production of IL-6, leading to bone destruction. In our study, NWT was observed to cause bone erosion while $\Delta nuc1$ did not. We assume the innate immune cells, such as macrophages and neutrophils, and other cell lines, such as osteoblasts and osteoclasts, may play a role in this process. Therefore, the SAOS-2 cells were used.

We have also explained clinical arthritis frequency and clinical arthritis score in Methods and Materials and Discussion as follows:

Line (96-97): The severity of septic arthritis was assessed by clinical arthritis frequency and clinical arthritis score.

Line (477-482): Clinical Evaluation of Arthritis

Observers (M.D. and T.J.) blinded to the treatment groups visually inspected all 4 limbs of each mouse. Arthritis was defined as erythema and/or swelling of the joints. A clinical scoring system ranging from 0 to 3 was used for each paw (0, no inflammation; 1, mild visible swelling and/or erythema; 2, moderate swelling and/or erythema; and 3, marked swelling and/or erythema).

2. The authors switch back and forth between strains. Primary mouse experiments are done in Newman, the next few *in vitro* experiments switch to SA113 and then to JE2 (USA300). The justification for this is completely lost on the reader and should be explained somewhere other than the discussion.

Thanks for pointing it out. We switched from SA113 Δ *igt* to JE2 Δ *igt* in the DNA degradation experiments. In the revised manuscript, we only focus on USA300 JE2 and Newman strains as they show similar pathogenicity and growth kinetics but USA300 JE2 is additionally resistant to methicillin.

We have updated the text as follows:

Line (172-175): Previously, we showed that the *S. aureus* USA300 JE2 has a similar virulence and pathogenicity pattern to NWT, but is additionally resistant to methicillin. Particularly, JE2 showed similar growth kinetics and hemolytic activity as NWT²⁰, and was therefore selected for further *in vitro* assays.

Line (312-315): To check the potential involvement of bacterial DNA and the role of NucA in inflammation, gDNA was isolated from the JE2 Δ *igt* mutant to avoid lipoprotein contamination. The NucA-digested DNA and intact DNA were then incubated with RAW 264.7 cells.

3. The authors pick 4 cytokines to measure in Figure 2. There is no clear explanation of why these were chosen. These cytokines change in Figure 4. There is once again no justification as to why this panel changed.

We have updated the text in the discussion about the cytokines as follows:

Line (346-371): To investigate this, we measured the levels of several cytokines in the blood of mice at 7-day post-infection. Previous studies have shown that IL-6 and TNF are critical for septic arthritis development^{32,38}. Patients who have septic arthritis exhibit high concentrations of TNF^{39,40}. Lack of TNF was associated with inefficient ability to clear bacteria³². KC attracts neutrophils that control septic arthritis, and S100A8/A9 has been identified as an early predictor of septic arthritis during *S. aureus* bacteremia^{4,41}. Although the *in vivo* experiments revealed little differences in TNF- α induction in response to the parental and the mutant, we decided to verify this observation also in cell lines. Consistent with the *in vivo* data, we observed no notable differences in TNF- α production by macrophage-like and osteoblast-like cells when incubated with JE2 and JE2 Δ *nuc1* (Figure 4a, b and Supplementary Figure 3). However, in neutrophils more TNF- α production was induced by the parental strain than the *nuc1* mutant. In the septic arthritis mouse model, NWT induced an almost 100-fold higher IL-6 production in plasma than its Δ *nuc1* mutant. One reason for the increased severity of septic arthritis upon infection with NucA-expressing strains could be the increased IL-6 content. Indeed, the proinflammatory IL-6, which is mainly produced by macrophages and T lymphocytes in response to pathogens, is not only a key player in rheumatoid arthritis. It also promotes megakaryocyte maturation and the release of platelets when reaching the bone marrow^{42,43}. The massive upregulation of S100A8/A9 in NWT-infected mice compared to the Δ *nuc1*-infected mice could be induced by IL-6 triggered inflammation and leukocyte recruitment^{44,45}. The high levels of IL-6 and S100A8/A9 at the end of the mouse experiment reflect the severity of infection. Inflammatory cytokines play a role in the bone remodeling process. For example, in IL-6 deficient mice, bone erosion was reduced⁴⁶. Therefore, the NucA-expressing strain could cause high IL-6 production, which may lead to uncontrolled progression of bone destruction by osteoclasts.

4. Line 122- 123 starts this section by explaining the role of TLR-9. TLR9 however never makes another appearance in the manuscript, other than in the discussion where it also feels completely disconnected.

We have added more to explain the connection as follows:

Line (148-153): *S. aureus* produces various virulent factors and triggers inflammation. Previous research found that the DNA from *S. aureus*, containing nonmethylated CpG motifs, acts as a factor that triggers arthritis and septic shock^{14,15}. Unmethylated bacterial DNA and CpG motifs are recognized by the toll-like receptor 9 (TLR9), which is expressed in immune cells such as macrophages and dendritic cells^{16,17}. Recognition via TLR9 pathway initiates the host response to *S. aureus* infection.

Line (310-312): It is known that bacterial DNA triggers an inflammatory response and induces cytokine production via TLR9 receptor. Further studies in mice have proven that DNA from *S. aureus* results in arthritis.

5. It is difficult to reconcile Figure 1d with 1e. While there is a large decrease in the abscess score for the nuclease mutant this translates to a less than 1 log drop of cfu. Images of kidneys to show abscesses would assist the interpretation of this data. Authors should consider additionally checking the synovial fluid for CFU.

Figure 1d showed the bacterial load in the kidney. In the $\Delta nuc1$ mutant there were 3 individuals with more than 2 log drops of cfu and 2 individuals with 1 log drop of cfu. We have updated this figure to log/cfu to make it clearer.

It is a good suggestion to check the CFU in the synovial fluid and show the abscesses figures. However, due to time constraints, it is challenging to complete this analysis here. The reduced bacterial burden in kidney and abscess score is also an indication for the decreased pathogenicity of $\Delta nuc1$.

6. Line 57-58 mentions biofilms but has no relevance to the rest of the manuscript. NucA has been demonstrated to break down NETs that are induced by *S. aureus* biofilms. These phenotypes/studies should be addressed especially since septic arthritis is associated with biofilm formation.

We have added more to explain the role of NucA in biofilm and NETs as follows:

Line (59-62): Additionally, neutrophils displayed higher NET formation when exposed to biofilms from the *nuc1* null mutant, and killed more bacterial cells, suggesting NucA is essential for the survival of *S. aureus* biofilms irrespective of whether the bacterium is phagocytosed or not^{7,8}.

7. Studies demonstrate that *S. aureus* can kill primary human neutrophils as quickly as 30 minutes following incubation. It is unclear how many neutrophils are alive at 2 hours or whether neutrophils are still alive at 5 hours post incubation. DNA can be released passively from neutrophils at this point. An earlier time point might provide clearer differences between strains, for NET release. The source of the Sytox labelled DNA is also unclear (PMN/bacterial). Authors should consider measuring MPO, neutrophil elastase and citrullinated histones as a measure of NETosis. Labelling of neutrophils with Cell Tracker would facilitate interpretation.

We have included the live (green)/dead (red) staining assay to check the survival of neutrophils after 2 h. In supplementary figure 5 we now show that most neutrophils incubated with live bacteria or supernatant display green staining, indicating they are still alive after 2 h.

We have also checked NET formation after 1h. Sytox-Green staining and immunofluorescence assays were performed to assess DNA abundance and origin in NETs. We observed a tendency of higher DNA abundance upon exposure to mutants as compared to *nuc1*-containing strains.

Immunofluorescence staining of NET formation (Figure 6b) further confirmed NETosis occurred as visible by positive staining for DNA (blue), myeloperoxidase (MPO, Green) and citrullinated histone H3 (CitH3, red) after 1h incubation of neutrophils with bacterial cell-free supernatant but not with live bacteria. This suggests that the supernatant of Δ *nuc1* mutants is less effective in degrading DNA, resulting in increased levels of NETs.

We also agree that cell tracker would be optimal to label the neutrophils when exposed to bacteria. Due to the time limitation, we did not perform this additional staining, but the live/dead staining and additional immunofluorescence assays clearly indicate that eDNA is released by neutrophils during NETosis and not as a result of lysis.

8. Previous studies demonstrate multiple phenotypes associated with deoxyadenosine generated following NET release, yet this not clearly addressed here. Does the release of neutrophil extracellular traps cause macrophage cytotoxicity as previously described? Considering previous studies were also performed in an intravenous model of sepsis, it would be important to exemplify the differences (and similarities) observed with some of these phenotypes.

We agreed that deoxyadenosine generation following NET release could be one of the reasons for this septic arthritis. We mention in discussion:

Line (392-397): It has been shown that the combined activity of *S. aureus* nuclease and adenosine synthase converts extracellular DNA to deoxyadenosine (dAdo) in staphylococcal abscesses. Human equilibrative nucleoside transporter 1 (hENT1) mediates dAdo transportation in macrophages, leading to caspase-3-induced apoptosis^{12,51,52}. This could be happening in our model as well, but we did not analyse the viability of macrophages isolated from abscesses.

9. Since a number of surface proteins have been demonstrated to contribute to virulence in septic models, it is especially relevant that isogenic mutants used in this manuscript be complemented in more than one assay.

We have now performed experiments with JE2, JE Δ *nuc1* and complemented strain JE2 Δ *nuc1*(pRB473-*nuc1*) to confirm the role of NucA. The JE2 Δ *nuc1*(pRB473-*nuc1*) formed a dark blue hole in DNase Test agar indicating the nuclease activity is restored. (Supplementary Figure 1d). Supplementary Figure 4 shows that both phagocytosis and bacterial survival were restored in the complemented strains when exposed to neutrophils. Additionally, the complemented strain showed similar internalization characteristics as the parent strain in RAW 264.7 and SAOS-2 cells.

We have also included immunofluorescence microscopy studies with the complemented strain, which triggered NET formation as the wild type (Fig. 6). Those results indicate that it is NucA to be responsible for internalization and induction of NET formation.

10. Is Figure 2a an n=1? Differences in Figure 2c are very difficult to see. Consider a zoom in box to the ROI. Since the novelty here, compared to previous studies, is the bone erosion phenotype, this should be clearly depicted. Images of complemented strain infection would solidify this further, especially since Newman is an osteomyelitis isolate.

Figure 2a illustrates the frequency of bone erosion (the percentage of eroded joints). We have removed the unnecessary dots. We agree with the reviewer that the current version of Figure 2c makes it hard to distinguish differences. To improve clarity, we have included larger images from 3D μ CT scans that clearly show the distinction between healthy and eroded joints.

11. MOI is 1:2 (presumably) but stated as MOI:2. It is also unclear whether neutrophils are 1 or 2. The methods state an MOI of 0.1 for neutrophil assays. The choice and reasons for change should be stated.

The MOI was 1:2 with neutrophils at 1. We have updated the discussion for different MOI.

Line (230-233): We determined bacterial survival and phagocytosis in neutrophils, following a previously used protocol^{27,28}. We assume a lower MOI mirrors the in vivo situation, while a higher MOI was used for the phagocytosis assay since it provides a clearer readout.

12. The rationale for Figure 5 is unclear. What is the difference between a killing assay and an uptake assay and why was uptake not performed in neutrophils or killing in macrophages/SAOS cells? Uptake cannot be concluded from this data unless it is clear that RAW and SAOS cells do not kill bacteria.

For the killing assay, we compared the CFU of *S. aureus* before and after incubating with neutrophils; and then we determined the survival as the ratio between CFU after and before incubation. To minimize the individual difference, we normalized it to JE2. When it comes to the uptake assay, we washed the extracellular bacteria and killed the adherent bacteria with lysostaphin and gentamycin in the macrophages/SAOS cells. Macrophages cells were lysed, and the bacteria plated on rich medium. After incubation, the internalized CFU of *S. aureus* was calculated.

We agree that discriminating between decreased internalization or increased killing in neutrophils could give us more valuable information. We have now performed a time-course survival assay in neutrophils, which revealed that JE2 Δ nuc1 exhibited decreased survival as compared to the wild type strain already 10 minutes after exposure to neutrophils (revised Fig. 5a), suggesting that its decreased internalization 1h after exposure to neutrophils likely results from increased killing. Similar results were obtained in RAW 264.7.

13. For Figure 5, it is unclear how these CFUs are calculated. The methods mention plating (assuming invasion assay is the uptake assay), but the axis says CFU/cell. An

appropriate measurement would be percent of time 0.
Line 173 mentions a 2-hour incubation, the legend says 1.5 hrs.

We have updated this to clarify the invasion assay.

Line (237-241): To assess only intracellular survival, membrane adherent and extracellular bacteria were killed after 1.5 h incubation, and then the CFU of internalized bacteria per host cell was determined. In RAW 264.7 we observed less intracellular numbers of JE2 Δ *nuc1* cells as compared to JE2 and complemented strain JE2 Δ *nuc1* (pRB*nuc1*), suggesting that survival was affected by *nuc1* (**Fig. 5b**).

Line (588-589): Internalization was calculated as CFU of internalized bacteria/host cell seeded and normalized to JE2.

The incubation time was 1.5 h. We have changed it.

14. Line 239- 242- Conclusions made here are far reaching and should be re-phrased. The authors conclude that NucA protects *S. aureus* from 1) engulfment- this is not shown in this manuscript and 2) killing- at the time point shown for neutrophils, only 20% of WT USA300 survives. That's not much of an advantage and could presumably reduce if left longer. Additionally, NucA degrading NETs is published, it is not a novel suggestion from data in this manuscript (Thammavongsa et al, 2013). This should be re-phrased.

We have updated some conclusions to make it clearer:

Line (290-297): Here, we provide three lines of *in vitro* evidence to support our *in vivo* observations: 1) NucA efficiently degrades bacterial DNA, which can trigger the release of proinflammatory cytokines like TNF- α , known for their critical roles in septic arthritis development; 2) NucA production may increase intracellular survival of *S. aureus*; 3) NucA produced by *S. aureus* effectively digests NETs formed by neutrophils, potentially aiding bacterial evasion from innate immune killing, and thereby promoting bacterial survival and exacerbating disease severity (Supplementary Figure 5).

It is true that several publications have shown that NucA degrades NETs and that the data on NETs are therefore not entirely new. However, the highlight of the work is the NucA-induced bone erosion in the mouse model of septic arthritis. To better understand this, we believe that we also need to investigate immune cells such as macrophages and neutrophils in this context, although there is already published data on the subject. The next task will be to investigate the mechanism underlying NucA-induced bone erosion. The present work is the basis for this follow-up study.

Minor Comments

1. Lines 192- 202 are very confusing and very difficult to interpret.

We have re-phrased it.

2. There are numerous typos in the manuscript such as line 31, line 81, line 89 (either/or not both/or), line 150, supplement line 9, line 300, line 302, line 305

We have corrected it.

3. Is Figure 2a an n=1? Differences in Figure 2c are very difficult to see.

We have changed the Figure 2c.

Point-by-point response to the referees' comments

Reviewer #1 (Remarks to the Author):

Thank you for the opportunity to review this paper titled “Staphylococcus aureus thermonuclease NucA is a key virulence factor in septic arthritis”. The manuscript reports that in the mouse model for hematogenous septic arthritis 23 the Δ nuc1 mutant was much less pathogenic and the severity of clinical septic arthritis 24 was markedly reduced, including decreased weight loss, lower kidney bacterial loads 25 and much less IL-6 production. Additionally, the manuscript demonstrates that NucA induced higher IL-6 production in SAOS-2 and higher TNF-28 α and IL-10 production in neutrophils and shielded S. aureus from phagocyte 29 engulfment and killing. This is an important area of investigation as septic arthritis is an aggressive joint disease causing significant morbidity and mortality. The authors were highly responsive to my previous suggestions and have incorporated sufficient additional data and information to alleviate all of my previous concerns.

We are grateful for your positive feedback. We also thank for the careful review.

Reviewer #2 (Remarks to the Author):

No further comments, the investigators addressed my comments appropriately. Outstanding work.

We appreciate for the encouraging comments and recognizing the value of our work.

Reviewer #3 (Remarks to the Author):

This manuscript demonstrates the role of the nuclease nuc1, as a virulence factor in a murine model of septic arthritis caused by S. aureus. I feel that this manuscript adds to our knowledge of the pathogenesis of S. aureus and will be valuable to the field. However, the writing would greatly benefit from an in-depth review of grammatical errors which includes numerous spelling errors and more importantly, sentence framing. These errors take away from the value of the data presented in this story.

For example, line 352- 354- Although the in vivo experiments revealed little differences in TNF- α induction in response to the parental and the mutant, we decided to verify this observation also in cell lines.

Suggestion- While in vivo experiments showed minimal differences in TNF-a

induction following infection with either WT or the isogenic nuclease mutant, we further evaluated these phenotypes *in vitro* and found that...

We appreciate your suggestions regarding sentence structure and grammatical errors. We have carefully revised this manuscript, corrected spelling and grammatical mistakes to enhance clarity and readability.

We also have formulated this sentence as follows.

Line (385-389): Since *in vivo* experiments showed minimal differences in TNF- α induction after infection with WT or *nuc1* mutant, we further investigated this phenotype *in vitro*. However, in macrophage-like and osteoblast-like cells, JE2 and JE2 Δ *nuc1* did not cause any significant difference in TNF- α production either (Figure 4a, b and Supplementary Figure 3).

Another example: Line 361- 363- Sentence abruptly ends.

We have updated it as follows:

Line (392-404): The very high IL-6 content in the blood reflects the severity of septic arthritis by infection with the NucA-expressing strain. In fact, sepsis or acute respiratory distress syndrome is correlated with an increased IL-6 content. However, there is no significant difference in IL-6 production between JE2 and its Δ *nuc1* mutant in the different cell cultures, showing once again that the *in vitro* situation does not always reflect the *in vivo* situation. IL-6, which is mainly produced by macrophages and T lymphocytes in response to pathogens, is not only a key player in rheumatoid arthritis, but it also promotes megakaryocyte maturation and the release of platelets when reaching the bone marrow. IL-6 has emerged alongside IL-1 and TNF as a master regulator of inflammation: it is essential for innate and adaptive immunity, is required for efficient pathogen clearance, and has important physiological roles in humans regulating the acute-phase response, hematopoiesis, metabolic rate, lipid homeostasis, and neural development.

Line 27-28- Typo. 'In vitro, *S. aureus* genomic DNA 27 induced a robust TNF- α response in macrophage-like cells was? abrogated when the DNA 28 was degraded by NucA'.

We have updated it.

Line (26-28): *In vitro*, *S. aureus* genomic DNA induced a robust TNF- α response in macrophage-like RAW 264.7 cells abrogated when the DNA was degraded by NucA.

Line 29- Respectively spelled wrong

We have corrected it.

Line 97- M&M should not be used in a manuscript in place of full words.

Figure 1 legend should be re-phrased. 'More clinical arthritis' should be replaced with increased severity or something similar.

I still think Figure 1d would benefit from images of kidney abscesses and that these should always be shown but if time is an issue, that's fine.

We have corrected the words and updated the figure.

Line (98-99): The severity of septic arthritis was assessed by clinical arthritis frequency and clinical arthritis score (see Materials and Methods for details).

Line (120-121): *S. aureus* Newman wild-type strain (NWT) imparts more severe arthritis and virulence than its $\Delta nuc1$ mutant during infection in NMRI mice.

We have also added the images of the kidney abscess in the Fig. 1f. This image clearly shows impressively the differences in kidney abscesses between NWT and $\Delta nuc1$ infected mice.

Line 124. Consider removing the word massive

We have replaced massive with significant.

Line 126- 127. The reasoning for this selection of cytokines needs to be introduced. One could argue for a number of other pro-inflammatory cytokines such as IL-1b, GCSF and GMCSF to be measured in the absence of this reasoning. Mentioning this in the discussion does not aid the interpretation of the results.

Thanks for point it out and we have added more introductions.

Line (129-134): IL-6 and TNF- α are essential for septic arthritis development. S100A8/A9 serves as a predictor of septic arthritis in bacteremic mice, and KC (CXCL1) recruits neutrophils, key innate immune cells essential for disease control. Accordingly, on day 7 post-infection we collected blood samples from the infected mice and measured the levels of these immune mediators: IL-6, TNF- α , KC (CXCL1), and S100A8/A9.

Line 148- Virulence

We have corrected it.

Line 158- The reader is not aware of what JE2 is, furthermore the correct annotation for the first time this strain is introduced, is USA300 JE2. (Mentioned later in line 172 and then switches back).

We have corrected it when it is first introduced.

Line (161-172): As staphylococcal macromolecules are frequently contaminated with lipoproteins/lipopeptides that are sensitively detected by Toll-like receptor 2 (TLR2) at picomolar levels, cytokine induction could be due to these constituents. Therefore, the mutant *S. aureus* USA300 LAC JE2 Δlgt was tested here as a control to recognize a possible interference of TLR2 and TLR9 ligands on cytokine production. JE2 is a plasmid-cured derivative of USA300 LAC strain, that is still methicillin-resistant and an important model strain to study *S. aureus* virulence. JE2 Δlgt lacks the phosphatidylglycerol:prolipoprotein *diacylglycerol transferase* Lgt, and therefore no lipidation of lipoproteins takes place and no TLR2 response can be triggered by

this mutant. By including this mutant and the double mutant JE2 Δ nuc1 Δ lgt together with JE2 and JE2 Δ nuc1 in the comparative immunostimulation studies, it is possible to specifically detect NucA-induced cytokine induction.

Line 172- 173. This is an overstated sentence. There are many studies where USA300 and Newman do not align on virulence properties, and the results depend on the infection context. This should be stated here.

We agree that our previous description about USA300 and Newman was overstated. We have revised the sentence and added the references.

Line (184-190): Comparative virulence studies in different *S. aureus* model strains have shown that the USA300 and Newman strains not only have comparable lethality in a mouse sepsis model but have also similar in their hemolytic activity and biofilm formation⁶. However, in other studies have shown that the two strains exhibit distinct virulence properties; their pathogenicity differs depending on the infection model. Both strains also express NucA and the corresponding Δ nuc1 mutants could be complemented by pRB473-nuc1 (**Supplementary Figure 1d**).

Line 175- 177. It seems like the introduction of the two types of nucleases is relevant at the beginning of the paper where the authors first mention nuc1 as the virulence factor of interest.

We fully agree and have put it into Introduction part.

Line (47-49): Of the two nucleases, it is the NucA that plays the crucial role in degrading extracellular DNA and RNA (eDNA and eRNA, respectively).

Line 199- Typo. Cytokine not cytokines

We have corrected it.

Line 207- This is the first time these cytokines are introduced. The equivalent of KC would be IL-8. IL-10 is not classically pro-inflammatory. The selection of IL-1Ra is unclear to me. The discussion provides a very detailed explanation of TNF- α and IL-6 but still doesn't tell me about these other cytokines. Additionally, introducing the context at the end of the paper diminishes its value for data interpretation. The revised text in the manuscript just says TNF. That's not a cytokine.

Thank you for your thoughtful thinking. We agree that the choice of immune mediators presented in this section differs from those highlighted in Fig. 2. *In vivo* assay (Fig. 2), we focused on immune mediators that are known to be produced in the septic arthritis model. However, in this particular section for *in vitro* assay, we utilized *in vitro* stimulation of macrophages, osteoblasts, and neutrophils. Consequently, the selection of immune mediators was guided by the molecules most relevant to the activation and response patterns of these cell types under *in vitro* conditions. We acknowledge that IL-10 is anti-inflammatory rather than classically pro-inflammatory. However, its crucial role in regulating and balancing inflammation makes it highly relevant in the context of infection and immune response, as IL-10

levels typically increase during inflammation. Similarly, the selection of IL-1Ra was intended to explore counter-regulatory mechanisms that could influence the inflammatory response. As a natural antagonist of IL-1 signaling, IL-1Ra provides valuable insights into the interplay between pro-inflammatory and anti-inflammatory mediators within this system.

We have expanded our introductions and discussion about cytokines selection and also corrected the TNF to TNF- α .

Line (218-222): To assess the immune responses elicited by different bacterial strains, we measured cytokine production, including classically pro-inflammatory mediators such as TNF- α and IL-6, as well as anti-inflammatory cytokines such as IL-10 and IL-1Ra. IL-10 was also very high in sepsis patients and IL-1Ra and IL-10 also influence the inflammatory response.

Line (365-376): In addition to measuring pro-inflammatory cytokines, we also measured the production of anti-inflammatory cytokines, IL-10 and IL-1Ra, in neutrophils to assess the balance between pro- and anti-inflammatory responses during infection. IL-10 is critical for regulating and balancing the infection and immune response, which typically increase during inflammation. IL-1Ra, as a natural antagonist of IL-1 signaling and related to extensive IL-10 production, could influence the inflammatory response and is used to treat inflammation. IL-1R knock-out mice developed more severe septic arthritis but IL-1R treatment for inflammation-related diseases could also increase the infections in sepsis. When neutrophils were exposed to parental strain and its mutants, JE2 Δ nuc1 showed less IL-10 and IL-1Ra production. Those results give a hint that NucA has an impact on immune stimulation.

Line 212-218. These are very strong assumptions made about the relative virulence differences between Newman and USA300 that are not supported by the data. Cytokine readouts (especially choosing 3) can be similar for many reasons and can elicit vastly different local and systemic responses during infection. Considering these macrophage and bone cells are not primary and that these phenotypes are not genetically complemented, the language of these statement is too strong and needs to be dialed back.

Thanks for pointing it out. We have now removed it and updated the text.

Line (227-234): It is not unexpected that the Δ lgt mutants induce less cytokines since lipoproteins/lipopeptides trigger a very strong innate immune response. As compared with NWT, there was less IL-6 production in RAW 264.7 and no significant difference in IL-6 and TNF- α in SAOS-2 when exposed to Δ nuc1 strain (**Supplementary Figure 3**). As the generated mutants showed hardly any difference in growth and hemolytic activity as compared to the parental strain (**Supplementary Figure 1b-d**), we assume that all the effects seen are mainly due to the deletion of *nuc1* or *lgt* genes.

Line 231- 233. What is this assumption based on? Literature? Reference required.

We have this assumption based on previous paper but overstating the conclusion. Now, we have revised it to ensure it accurately reflects the existing literature.

Line (246-249): We determined bacterial survival and phagocytosis in neutrophils, following a previously used protocol. With different conditions tested, we found that better results were obtained with an MOI=2 for the phagocytosis assay and an MOI=0.1 for the bacterial killing assay.

Line 244- The complementation was made on an episome?

Yes. The complementary strain was made with a plasmid, pRB473-*nuc1*.

Figure 5 is very confusing. The legend mentions that RFU are plotted in A, but of what, is completely lost. The red anti-citrullinated histone stain is not easy to see, the channels should be separated in order for the red channel to be highlighted. The novelty of these findings is also unclear <https://pmc.ncbi.nlm.nih.gov/articles/PMC2982853/>. Explaining the reasoning for these experiments in the context of this story, at the beginning of this section would assist the manuscript.

We have updated the figure legend 6 to explain what the RFU is.

Line (296-297): Relative fluorescence units (RFU) of Sytox Green normalized to Triton X-100-lysed neutrophils are shown.

We also agree that the red dots are not so clear and have now separated them into different channels. The updated images are shown in Supplementary Figure 5 with DNA, MPO and citH3 channels separated.

We have also added more introductions in this context.

Line (274-278): Neutrophils act as the first defense line of the innate immune system and form NETs to clear pathogens. NETs consist of a DNA backbone coated with various proteins, such as myeloperoxidase (MPO), nuclear proteins (histones), neutrophil elastase (NE), and calprotectin. In *S. aureus*, NucA can degrade extracellular DNA, thereby reducing NET formation and evading immune clearance.

Line 276. Representative images should be demonstrated, even when not exemplary.

We have updated representative images in Fig. 6b.

Line 285- The word massive is not scientific or specific and should not be used to describe data.

We have replaced the massive to marked, severe or pronounced.

Line 310- 318. If TLR-9 is the only thing that initiates a host response to *S. aureus*

please provide references of this statement. This entire paragraph has no references.

Not only the TLR9 pathway but also other pathways could induce septic arthritis. We have now added relevant references to state and ensure the entire paragraph is well-documented.

Line 330- 336. Are there any references for this paragraph?

We have included the reference and updated the text.

Line (353-356): As we showed that NucA-digested bacterial eDNA has no immune-stimulating activity in murine macrophages, we expected that the $\Delta nuc1$ mutant, in which eDNA is not degraded (**Supplementary Figure 1a**), would elicit a stronger immune response than the parental strain in various cell types.

Line 527- Typo. Neutrophil not neutrophils

We have corrected it.

Supplement lines 28-29. Grammar

We have corrected it.

Point-by-point response to reviewers

Reviewer #1 (Remarks to the Author):

Thank you for the opportunity to review this paper titled “Staphylococcus aureus thermonuclease NucA is a key virulence factor in septic arthritis”. The manuscript reports that in the mouse model for hematogenous septic arthritis 23 the $\Delta nuc1$ mutant was much less pathogenic and the severity of clinical septic arthritis 24 was markedly reduced, including decreased weight loss, lower kidney bacterial loads 25 and much less IL-6 production. Additionally, the manuscript demonstrates that NucA induced higher IL-6 production in SAOS-2 and higher TNF-28 α and IL-10 production in neutrophils and shielded *S. aureus* from phagocyte 29 engulfment and killing. This is an important area of investigation as septic arthritis is an aggressive joint disease causing significant morbidity and mortality. Below I have suggested some modifications and additional experiments to strengthen the manuscript. I wish you the best with this submission.

1. The manuscript would benefit from further discussion of the rationale for focusing on TNF in later figures given that Figure 2D demonstrates minimal differences in TNF production between NWT and the *nuc1* deletion mutant.

Thank you for pointing this out. Although the *in vivo* experiments revealed little differences in TNF- α induction in response to the parental and the mutant, we wanted to verify this observation also in cell lines because this cytokine is strongly released by macrophages in response to conserved bacterial components such as peptidoglycan, lipoprotein, protein A and DNA.

We have included the text in the discussion about the TNF cytokines as follows:
Line (352-358): Although the *in vivo* experiments revealed little differences in TNF- α induction in response to the parental and the mutant, we decided to verify this observation also in cell lines. Consistent with the *in vivo* data, we observed no notable differences in TNF- α production by macrophage-like and osteoblast-like cells when incubated with JE2 and JE2 $\Delta nuc1$ (Figure 4a, b and Supplementary Figure 3). However, in neutrophils more TNF- α production was induced by the parental strain than the *nuc1* mutant.

2. In the discussion of Figure 3 the manuscript states that both ODN2006 and ODN2137 were employed in panel 3A (Lines 134-136). However, the data for ODN2137 was not included and is an important negative control.

Thanks for pointing this out. In this study, standard CpG oligonucleotides ODN2006 and the non-CpG oligonucleotides ODN2137 were employed as positive and negative controls, respectively, to illustrate bacterial DNA-induced release of TNF- α in RAW 264.7 cells. But we forgot to include the negative control ODN2137 in Fig. 3. We apologize for overlooking this. We now include ODN2137 in the revised version of Fig. 3a.

3. The manuscript would be strengthened from documenting experimentally or referencing previous characterization of the strains employed (JE2, JE2 $\Delta nuc1$,

JE2 Δ lgt, JE2 Δ nuc1 Δ lgt) in terms of nuc1 released by bacteria and capacity to degrade DNA. Additionally, it is necessary to characterize the presence of extracellular DNA available for nuc1 digestion. Currently, the data in figure 4 could be alternatively explained by differences in nuc1 release, nuc1 activity, or differences in available extracellular DNA during infection of RAW 264.7, SAOS-2, and Neutrophils. In particular, for infection of neutrophils are NETs formed for this MOI at this time point?

We have updated the text as follows to illustrate the difference in NucA activity across the used strains.

Line (175-183): Of the two nuclease genes (*nuc1* and *nuc2*) in *S. aureus*, it is the *nuc1*-encoded NucA that is secreted into the supernatant and is able to degrade extracellular DNA and RNA (eDNA and eRNA, respectively)²¹. This was demonstrated when we incubated the supernatant of JE2 and its mutants JE2 Δ nuc1, JE2 Δ lgt, and JE2 Δ nuc1 Δ lgt with *S. aureus* gDNA for 1 h to evaluate the nuclease activity. In all mutants in which *nuc1* was deleted, the gDNA remained intact, whereas in JE2 and the JE2 Δ lgt mutant the gDNA was completely degraded, indicating that NucA is responsible for gDNA degradation (Supplementary Figure 1a).

The question as to differences in available extracellular DNA during infection of RAW 264.7, SAOS-2, and neutrophils is difficult to understand because the reviewer refers to Figure 4, where we show cytokine quantification in response to the various bacterial strains. We quantify NET formation and degradation in neutrophils only (this is figure 6) by fluorescently staining the DNA in NETs, which is shown in blue in the picture. The supernatants from JE2 Δ nuc1 and JE2 Δ nuc1 Δ lgt triggered more eDNA as compared to JE2 and JE2 Δ lgt, indicating that NucA is responsible for eDNA degradation. In this experiment, we used a MOI of 2, which induces very little NETs formation after 1 h (Fig. 6c).

4. The manuscript examines bacterial killing in various cell types in Figure 5. In particular the data indicate a significant reduction in bacterial survival in neutrophils for the JE2 Δ nuc1, JE2 Δ lgt, JE2 Δ nuc1 Δ lgt strains compared to JE2. The manuscript indicates this is due to differences in internalization. However, it does examine internalization in the context of neutrophils only RAW 264.7 and SAOS-2.

We agree that investigating the internalization in neutrophils could give us more valuable information. We have now performed a time-course survival assay in neutrophils, which revealed that JE2 Δ nuc1 exhibited decreased survival as compared to the wild type strain already 10 minutes after exposure to neutrophils (revised Fig. 5a), suggesting that its decreased internalization 1h after exposure to neutrophils likely results from increased killing. Similar results were obtained in RAW 264.7 cells.

5. In figure 5B-C presents data indicating reduced recovery of intracellular viable CFU for the JE2 Δ nuc1, JE2 Δ lgt, JE2 Δ nuc1 Δ lgt strains compared to JE2 suggesting a reduction in internalization. Additional immunofluorescence experiments should be conducted to confirm that this is in fact due to internalization. With the current experimental the mutants could display the same degree of internalization as the parent JE2 but simply be more susceptible to intracellular killing.

See the reply to point 4.

6. The manuscript states in the discussion that NucA shields *S. aureus* from phagocyte engulfment. Additional experiments should be included to fully demonstrate that there is reduced internalization with the JE2 Δ nuc1 strain compared to the parent JE2. It would further support this statement to complement the JE2 Δ nuc1 infection with nuclease to demonstrate that this can restore internalization and survival levels to that observed for the JE2 parent.

We have now added experiments with JE2, JE Δ nuc1 and complemented strain JE2 Δ nuc1(pRB473-nuc1) to confirm the role of NucA. The JE2 Δ nuc1(pRB473-nuc1) formed a dark blue hole in the DNase Test agar indicating that the nuclease ability was restored (Supplementary Figure 1d). Fig. 5 shows that the phagocytosis and bacterial survival were restored in the complemented strains when incubated with neutrophils. Additionally, the complemented Δ nuc1 mutant showed a similar internalization pattern as the parent strain in RAW 264.7.

7. The manuscript would benefit from a schematic documenting the authors' current results in the context of septic arthritis.

It is a good suggestion. We have included a schematic representation to explain our results about septic arthritis.

Reviewer #2 (Remarks to the Author):

This is a very interesting manuscript detailing clearly that NucA is a significant factor in *S. aureus* septic arthritis. This was documented using well described animal models of infection. It was interesting to note, however, that while NucA is also important in abscess formation in the kidney, it may not be important in metastasis through the vasculature or liver as the same bacterial burden was noted in the kidney.

It was also very clear that NucA functions to degrade staphylococcal DNA and in addition eDNA/RNA from the host. However, there were other aspects of the manuscript that seemed disjointed. First, it was unclear why the investigators utilized various cell lines in this manuscript--it was not clear to the reader why RAW 264.7 and SAOS-2 cells were used and how one can interpret the different cytokine responses that were noted.

Thanks for your thoughtful feedback. We agree that we did not clearly explain the rationale behind the use of the various cell lines. In our mouse model, NWT was observed to cause bone erosion, whereas Δ nuc1 did not. Therefore, we assume the innate immune cells, such as macrophages and neutrophils, and other cell lines, such as osteoblasts and osteoclasts, may play a role in the observed phenotype. Accordingly, macrophages RAW 264.7 and osteoblasts SAOS-2 were employed.

We have now included the explanation about the cell lines used in this manuscript as follows:

Line (200-203): *S. aureus* activates different host cells, including macrophages, neutrophils and osteoblasts²²⁻²⁴. We further investigated whether the immune stimulation of JE2 and its mutants differed in RAW 264.7, neutrophils and osteoblast-like SAOS-2 cells.

Line (333-337): *S. aureus* stimulates the immune response in various cells, including macrophages and neutrophils, as well as non-immune cells, such as osteoblasts cells. In addition, osteoblast and osteoclast cells are responsible for bone remodeling and construction. Accordingly, macrophages RAW 264.7, osteoblast cells SAOS-2 and neutrophils were employed in this study.

The differences in IL-6 production between RAW 264.7 and SAOS-2, with the former producing higher amounts, clearly reflect the important role of macrophages in the response to bacteria as compared to osteoblasts, which is supported by other studies.

Second, although it was discussed that one potential reason NucA is so important in septic arthritis is that it induces a significant IL-6 response, the investigators did not use an IL-6 knockout mouse to address this hypothesis. Unless there are other mitigating factors that are not understood, this seems to be an important point.

In the $\Delta nuc1$ -infected mice, largely decreased IL-6 and S100A8/A9 were observed. We totally agree that IL-6 plays a role in this process. Other groups have shown that a lack of IL-6 would lead to decreased arthritis scores and bone erosion. High concentration of IL-6 is observed in patients with septic arthritis. However, analyzing the IL-6 knockout mouse in the limited revision time we have available is not feasible. We will do it in the future.

Minor comments:

the authors might want to replace massive with significant or another word throughout the manuscript

We have replaced massive with significant, markable and notable.

I am not familiar with the term diffuse areal and was unable to find this term in the literature (or at least associated with biology)?

We have re-phased the whole paragraph and Figure 6 to make it clearer.

Unclear that the discussion from lines 272-280 are needed. In my opinion, no need to justify why another strain was not used.

We have deleted it and provided a brief mention (172-175) in Results.

Neutrophils instead of neutrophiles in line 31.

We have corrected it.

Reviewer #3 (Remarks to the Author):

Summary. In this manuscript authors demonstrate a role for the extracellular nuclease, NucA in the pathogenesis of *S. aureus* during septic arthritis. This was found to be associated with the ability of NucA to prevent the induction of an appropriate, inflammatory immune response to *S. aureus*. The authors demonstrate a very interesting phenotype for bone erosion caused when NucA is expressed in vivo, that may be related to the levels of inflammation induced in response to *S. aureus* DNA, providing exciting avenues for further research.

Key Strengths. To the best of my knowledge, this is the first report for the role of nucA in septic arthritis and control of local inflammation caused therein. The methods used in the manuscript are well established and robust.

Key Weaknesses. The writing is very confusing. Rationale for certain experiments (and conclusions made thereof) is unclear.

Major Comments.

1. The writing of this manuscript and the grammatical style is very difficult to follow, and tense uses make interpretation of the data tenuous. There are also a number of abbreviations used throughout the manuscript that are never introduced. What are interna mice? What are SAOS-2 cells and why are they relevant here? Additionally, an introduction of terms such as 'clinical arthritis frequency' and 'clinical arthritis score' would aid the interpretation of results.

We appreciate the constructive feedback and apologize for the grammatical errors. We have carefully revised the manuscript, especially the conclusion part, and added the explanation to abbreviations and relevance in cell lines and mouse model.

Naval Medical Research Institute (NMRI) mice are an outbred mouse strain with larger genetic variations, resembling the clinical situation. They were employed in our study.

SAOS-2 cells are osteoblast cells. Osteoblasts join in bone remodeling and immune response. Previous studies have investigated whether *S. aureus* could invade osteoblasts and induce production of IL-6, leading to bone destruction. In our study, NWT was observed to cause bone erosion while $\Delta nuc1$ did not. We assume the innate immune cells, such as macrophages and neutrophils, and other cell lines, such as osteoblasts and osteoclasts, may play a role in this process. Therefore, the SAOS-2 cells were used.

We have also explained clinical arthritis frequency and clinical arthritis score in Methods and Materials and Discussion as follows:

Line (96-97): The severity of septic arthritis was assessed by clinical arthritis frequency and clinical arthritis score.

Line (477-482): Clinical Evaluation of Arthritis

Observers (M.D. and T.J.) blinded to the treatment groups visually inspected all 4 limbs of each mouse. Arthritis was defined as erythema and/or swelling of the joints. A clinical scoring system ranging from 0 to 3 was used for each paw (0, no inflammation; 1, mild visible swelling and/or erythema; 2, moderate swelling and/or erythema; and 3, marked swelling and/or erythema).

2. The authors switch back and forth between strains. Primary mouse experiments are done in Newman, the next few *in vitro* experiments switch to SA113 and then to JE2 (USA300). The justification for this is completely lost on the reader and should be explained somewhere other than the discussion.

Thanks for pointing it out. We switched from SA113 Δ *igt* to JE2 Δ *igt* in the DNA degradation experiments. In the revised manuscript, we only focus on USA300 JE2 and Newman strains as they show similar pathogenicity and growth kinetics but USA300 JE2 is additionally resistant to methicillin.

We have updated the text as follows:

Line (172-175): Previously, we showed that the *S. aureus* USA300 JE2 has a similar virulence and pathogenicity pattern to NWT, but is additionally resistant to methicillin. Particularly, JE2 showed similar growth kinetics and hemolytic activity as NWT²⁰, and was therefore selected for further *in vitro* assays.

Line (312-315): To check the potential involvement of bacterial DNA and the role of NucA in inflammation, gDNA was isolated from the JE2 Δ *igt* mutant to avoid lipoprotein contamination. The NucA-digested DNA and intact DNA were then incubated with RAW 264.7 cells.

3. The authors pick 4 cytokines to measure in Figure 2. There is no clear explanation of why these were chosen. These cytokines change in Figure 4. There is once again no justification as to why this panel changed.

We have updated the text in the discussion about the cytokines as follows:

Line (346-371): To investigate this, we measured the levels of several cytokines in the blood of mice at 7-day post-infection. Previous studies have shown that IL-6 and TNF are critical for septic arthritis development^{32,38}. Patients who have septic arthritis exhibit high concentrations of TNF^{39,40}. Lack of TNF was associated with inefficient ability to clear bacteria³². KC attracts neutrophils that control septic arthritis, and S100A8/A9 has been identified as an early predictor of septic arthritis during *S. aureus* bacteremia^{4,41}. Although the *in vivo* experiments revealed little differences in TNF- α induction in response to the parental and the mutant, we decided to verify this observation also in cell lines. Consistent with the *in vivo* data, we observed no notable differences in TNF- α production by macrophage-like and osteoblast-like cells when incubated with JE2 and JE2 Δ *nuc1* (Figure 4a, b and Supplementary Figure 3). However, in neutrophils more TNF- α production was induced by the parental strain than the *nuc1* mutant. In the septic arthritis mouse model, NWT induced an almost 100-fold higher IL-6 production in plasma than its Δ *nuc1* mutant. One reason for the increased severity of septic arthritis upon infection with NucA-expressing strains could be the increased IL-6 content. Indeed, the proinflammatory IL-6, which is mainly produced by macrophages and T lymphocytes in response to pathogens, is not only a key player in rheumatoid arthritis. It also promotes megakaryocyte maturation and the release of platelets when reaching the bone marrow^{42,43}. The massive upregulation of S100A8/A9 in NWT-infected mice compared to the Δ *nuc1*-infected mice could be induced by IL-6 triggered inflammation and leukocyte recruitment^{44,45}. The high levels of IL-6 and S100A8/A9 at the end of the mouse experiment reflect the severity of infection. Inflammatory cytokines play a role in the bone remodeling process. For example, in IL-6 deficient mice, bone erosion was reduced⁴⁶. Therefore, the NucA-expressing strain could cause high IL-6 production, which may lead to uncontrolled progression of bone destruction by osteoclasts.

4. Line 122- 123 starts this section by explaining the role of TLR-9. TLR9 however never makes another appearance in the manuscript, other than in the discussion where it also feels completely disconnected.

We have added more to explain the connection as follows:

Line (148-153): *S. aureus* produces various virulent factors and triggers inflammation. Previous research found that the DNA from *S. aureus*, containing nonmethylated CpG motifs, acts as a factor that triggers arthritis and septic shock^{14,15}. Unmethylated bacterial DNA and CpG motifs are recognized by the toll-like receptor 9 (TLR9), which is expressed in immune cells such as macrophages and dendritic cells^{16,17}. Recognition via TLR9 pathway initiates the host response to *S. aureus* infection.

Line (310-312): It is known that bacterial DNA triggers an inflammatory response and induces cytokine production via TLR9 receptor. Further studies in mice have proven that DNA from *S. aureus* results in arthritis.

5. It is difficult to reconcile Figure 1d with 1e. While there is a large decrease in the abscess score for the nuclease mutant this translates to a less than 1 log drop of cfu. Images of kidneys to show abscesses would assist the interpretation of this data. Authors should consider additionally checking the synovial fluid for CFU.

Figure 1d showed the bacterial load in the kidney. In the $\Delta nuc1$ mutant there were 3 individuals with more than 2 log drops of cfu and 2 individuals with 1 log drop of cfu. We have updated this figure to log/cfu to make it clearer.

It is a good suggestion to check the CFU in the synovial fluid and show the abscesses figures. However, due to time constraints, it is challenging to complete this analysis here. The reduced bacterial burden in kidney and abscess score is also an indication for the decreased pathogenicity of $\Delta nuc1$.

6. Line 57-58 mentions biofilms but has no relevance to the rest of the manuscript. NucA has been demonstrated to break down NETs that are induced by *S. aureus* biofilms. These phenotypes/studies should be addressed especially since septic arthritis is associated with biofilm formation.

We have added more to explain the role of NucA in biofilm and NETs as follows:

Line (59-62): Additionally, neutrophils displayed higher NET formation when exposed to biofilms from the *nuc1* null mutant, and killed more bacterial cells, suggesting NucA is essential for the survival of *S. aureus* biofilms irrespective of whether the bacterium is phagocytosed or not^{7,8}.

7. Studies demonstrate that *S. aureus* can kill primary human neutrophils as quickly as 30 minutes following incubation. It is unclear how many neutrophils are alive at 2 hours or whether neutrophils are still alive at 5 hours post incubation. DNA can be released passively from neutrophils at this point. An earlier time point might provide clearer differences between strains, for NET release. The source of the Sytox labelled DNA is also unclear (PMN/bacterial). Authors should consider measuring MPO, neutrophil elastase and citrullinated histones as a measure of NETosis. Labelling of neutrophils with Cell Tracker would facilitate interpretation.

We have included the live (green)/dead (red) staining assay to check the survival of neutrophils after 2 h. In supplementary figure 5 we now show that most neutrophils incubated with live bacteria or supernatant display green staining, indicating they are still alive after 2 h.

We have also checked NET formation after 1h. Sytox-Green staining and immunofluorescence assays were performed to assess DNA abundance and origin in NETs. We observed a tendency of higher DNA abundance upon exposure to mutants as compared to *nuc1*-containing strains.

Immunofluorescence staining of NET formation (Figure 6b) further confirmed NETosis occurred as visible by positive staining for DNA (blue), myeloperoxidase (MPO, Green) and citrullinated histone H3 (CitH3, red) after 1h incubation of neutrophils with bacterial cell-free supernatant but not with live bacteria. This suggests that the supernatant of Δ *nuc1* mutants is less effective in degrading DNA, resulting in increased levels of NETs.

We also agree that cell tracker would be optimal to label the neutrophils when exposed to bacteria. Due to the time limitation, we did not perform this additional staining, but the live/dead staining and additional immunofluorescence assays clearly indicate that eDNA is released by neutrophils during NETosis and not as a result of lysis.

8. Previous studies demonstrate multiple phenotypes associated with deoxyadenosine generated following NET release, yet this not clearly addressed here. Does the release of neutrophil extracellular traps cause macrophage cytotoxicity as previously described? Considering previous studies were also performed in an intravenous model of sepsis, it would be important to exemplify the differences (and similarities) observed with some of these phenotypes.

We agreed that deoxyadenosine generation following NET release could be one of the reasons for this septic arthritis. We mention in discussion:

Line (392-397): It has been shown that the combined activity of *S. aureus* nuclease and adenosine synthase converts extracellular DNA to deoxyadenosine (dAdo) in staphylococcal abscesses. Human equilibrative nucleoside transporter 1 (hENT1) mediates dAdo transportation in macrophages, leading to caspase-3-induced apoptosis^{12,51,52}. This could be happening in our model as well, but we did not analyse the viability of macrophages isolated from abscesses.

9. Since a number of surface proteins have been demonstrated to contribute to virulence in septic models, it is especially relevant that isogenic mutants used in this manuscript be complemented in more than one assay.

We have now performed experiments with JE2, JE Δ *nuc1* and complemented strain JE2 Δ *nuc1*(pRB473-*nuc1*) to confirm the role of NucA. The JE2 Δ *nuc1*(pRB473-*nuc1*) formed a dark blue hole in DNase Test agar indicating the nuclease activity is restored. (Supplementary Figure 1d). Supplementary Figure 4 shows that both phagocytosis and bacterial survival were restored in the complemented strains when exposed to neutrophils. Additionally, the complemented strain showed similar internalization characteristics as the parent strain in RAW 264.7 and SAOS-2 cells.

We have also included immunofluorescence microscopy studies with the complemented strain, which triggered NET formation as the wild type (Fig. 6). Those results indicate that it is NucA to be responsible for internalization and induction of NET formation.

10. Is Figure 2a an n=1? Differences in Figure 2c are very difficult to see. Consider a zoom in box to the ROI. Since the novelty here, compared to previous studies, is the bone erosion phenotype, this should be clearly depicted. Images of complemented strain infection would solidify this further, especially since Newman is an osteomyelitis isolate.

Figure 2a illustrates the frequency of bone erosion (the percentage of eroded joints). We have removed the unnecessary dots. We agree with the reviewer that the current version of Figure 2c makes it hard to distinguish differences. To improve clarity, we have included larger images from 3D μ CT scans that clearly show the distinction between healthy and eroded joints.

11. MOI is 1:2 (presumably) but stated as MOI:2. It is also unclear whether neutrophils are 1 or 2. The methods state an MOI of 0.1 for neutrophil assays. The choice and reasons for change should be stated.

The MOI was 1:2 with neutrophils at 1. We have updated the discussion for different MOI.

Line (230-233): We determined bacterial survival and phagocytosis in neutrophils, following a previously used protocol^{27,28}. We assume a lower MOI mirrors the in vivo situation, while a higher MOI was used for the phagocytosis assay since it provides a clearer readout.

12. The rationale for Figure 5 is unclear. What is the difference between a killing assay and an uptake assay and why was uptake not performed in neutrophils or killing in macrophages/SAOS cells? Uptake cannot be concluded from this data unless it is clear that RAW and SAOS cells do not kill bacteria.

For the killing assay, we compared the CFU of *S. aureus* before and after incubating with neutrophils; and then we determined the survival as the ratio between CFU after and before incubation. To minimize the individual difference, we normalized it to JE2. When it comes to the uptake assay, we washed the extracellular bacteria and killed the adherent bacteria with lysostaphin and gentamycin in the macrophages/SAOS cells. Macrophages cells were lysed, and the bacteria plated on rich medium. After incubation, the internalized CFU of *S. aureus* was calculated.

We agree that discriminating between decreased internalization or increased killing in neutrophils could give us more valuable information. We have now performed a time-course survival assay in neutrophils, which revealed that JE2 Δ nuc1 exhibited decreased survival as compared to the wild type strain already 10 minutes after exposure to neutrophils (revised Fig. 5a), suggesting that its decreased internalization 1h after exposure to neutrophils likely results from increased killing. Similar results were obtained in RAW 264.7.

13. For Figure 5, it is unclear how these CFUs are calculated. The methods mention plating (assuming invasion assay is the uptake assay), but the axis says CFU/cell. An

appropriate measurement would be percent of time 0.
Line 173 mentions a 2-hour incubation, the legend says 1.5 hrs.

We have updated this to clarify the invasion assay.

Line (237-241): To assess only intracellular survival, membrane adherent and extracellular bacteria were killed after 1.5 h incubation, and then the CFU of internalized bacteria per host cell was determined. In RAW 264.7 we observed less intracellular numbers of JE2 Δ *nuc1* cells as compared to JE2 and complemented strain JE2 Δ *nuc1* (pRB*nuc1*), suggesting that survival was affected by *nuc1* (**Fig. 5b**).

Line (588-589): Internalization was calculated as CFU of internalized bacteria/host cell seeded and normalized to JE2.

The incubation time was 1.5 h. We have changed it.

14. Line 239- 242- Conclusions made here are far reaching and should be re-phrased. The authors conclude that NucA protects *S. aureus* from 1) engulfment- this is not shown in this manuscript and 2) killing- at the time point shown for neutrophils, only 20% of WT USA300 survives. That's not much of an advantage and could presumably reduce if left longer. Additionally, NucA degrading NETs is published, it is not a novel suggestion from data in this manuscript (Thammavongsa et al, 2013). This should be re-phrased.

We have updated some conclusions to make it clearer:

Line (290-297): Here, we provide three lines of *in vitro* evidence to support our *in vivo* observations: 1) NucA efficiently degrades bacterial DNA, which can trigger the release of proinflammatory cytokines like TNF- α , known for their critical roles in septic arthritis development; 2) NucA production may increase intracellular survival of *S. aureus*; 3) NucA produced by *S. aureus* effectively digests NETs formed by neutrophils, potentially aiding bacterial evasion from innate immune killing, and thereby promoting bacterial survival and exacerbating disease severity (Supplementary Figure 5).

It is true that several publications have shown that NucA degrades NETs and that the data on NETs are therefore not entirely new. However, the highlight of the work is the NucA-induced bone erosion in the mouse model of septic arthritis. To better understand this, we believe that we also need to investigate immune cells such as macrophages and neutrophils in this context, although there is already published data on the subject. The next task will be to investigate the mechanism underlying NucA-induced bone erosion. The present work is the basis for this follow-up study.

Minor Comments

1. Lines 192- 202 are very confusing and very difficult to interpret.

We have re-phrased it.

2. There are numerous typos in the manuscript such as line 31, line 81, line 89 (either/or not both/or), line 150, supplement line 9, line 300, line 302, line 305

We have corrected it.

3. Is Figure 2a an n=1? Differences in Figure 2c are very difficult to see.

We have changed the Figure 2c.

Point-by-point response to the referees' comments

Reviewer #1 (Remarks to the Author):

Thank you for the opportunity to review this paper titled “Staphylococcus aureus thermonuclease NucA is a key virulence factor in septic arthritis”. The manuscript reports that in the mouse model for hematogenous septic arthritis 23 the Δ nuc1 mutant was much less pathogenic and the severity of clinical septic arthritis 24 was markedly reduced, including decreased weight loss, lower kidney bacterial loads 25 and much less IL-6 production. Additionally, the manuscript demonstrates that NucA induced higher IL-6 production in SAOS-2 and higher TNF-28 α and IL-10 production in neutrophils and shielded *S. aureus* from phagocyte 29 engulfment and killing. This is an important area of investigation as septic arthritis is an aggressive joint disease causing significant morbidity and mortality. The authors were highly responsive to my previous suggestions and have incorporated sufficient additional data and information to alleviate all of my previous concerns.

We are grateful for your positive feedback. We also thank for the careful review.

Reviewer #2 (Remarks to the Author):

No further comments, the investigators addressed my comments appropriately. Outstanding work.

We appreciate for the encouraging comments and recognizing the value of our work.

Reviewer #3 (Remarks to the Author):

This manuscript demonstrates the role of the nuclease nuc1, as a virulence factor in a murine model of septic arthritis caused by *S. aureus*. I feel that this manuscript adds to our knowledge of the pathogenesis of *S. aureus* and will be valuable to the field. However, the writing would greatly benefit from an in-depth review of grammatical errors which includes numerous spelling errors and more importantly, sentence framing. These errors take away from the value of the data presented in this story.

For example, line 352- 354- Although the *in vivo* experiments revealed little differences in TNF- α induction in response to the parental and the mutant, we decided to verify this observation also in cell lines.

Suggestion- While *in vivo* experiments showed minimal differences in TNF- α induction following infection with either WT or the isogenic nuclease mutant, we further evaluated these phenotypes *in vitro* and found that...

We appreciate your suggestions regarding sentence structure and grammatical errors. We have carefully revised this manuscript, corrected spelling and grammatical mistakes to enhance clarity and readability.

We also have formulated this sentence as follows.

Line (385-389): Since *in vivo* experiments showed minimal differences in TNF- α induction after infection with WT or *nuc1* mutant, we further investigated this phenotype *in vitro*. However, in macrophage-like and osteoblast-like cells, JE2 and

JE2 Δ nuc1 did not cause any significant difference in TNF- α production either (Figure 4a, b and Supplementary Figure 3).

Another example: Line 361- 363- Sentence abruptly ends.

We have updated it as follows:

Line (392-404): The very high IL-6 content in the blood reflects the severity of septic arthritis by infection with the NucA-expressing strain. In fact, sepsis or acute respiratory distress syndrome is correlated with an increased IL-6 content. However, there is no significant difference in IL-6 production between JE2 and its Δ nuc1 mutant in the different cell cultures, showing once again that the *in vitro* situation does not always reflect the *in vivo* situation. IL-6, which is mainly produced by macrophages and T lymphocytes in response to pathogens, is not only a key player in rheumatoid arthritis, but it also promotes megakaryocyte maturation and the release of platelets when reaching the bone marrow. IL-6 has emerged alongside IL-1 and TNF as a master regulator of inflammation: it is essential for innate and adaptive immunity, is required for efficient pathogen clearance, and has important physiological roles in humans regulating the acute-phase response, hematopoiesis, metabolic rate, lipid homeostasis, and neural development.

Line 27-28- Typo. 'In vitro, *S. aureus* genomic DNA 27 induced a robust TNF- α response in macrophage-like cells was? abrogated when the DNA 28 was degraded by NucA'.

We have updated it.

Line (26-28): *In vitro*, *S. aureus* genomic DNA induced a robust TNF- α response in macrophage-like RAW 264.7 cells abrogated when the DNA was degraded by NucA.

Line 29- Respectively spelled wrong

We have corrected it.

Line 97- M&M should not be used in a manuscript in place of full words.

Figure 1 legend should be re-phrased. 'More clinical arthritis' should be replaced with increased severity or something similar.

I still think Figure 1d would benefit from images of kidney abscesses and that these should always be shown but if time is an issue, that's fine.

We have corrected the words and updated the figure.

Line (98-99): The severity of septic arthritis was assessed by clinical arthritis frequency and clinical arthritis score (see Materials and Methods for details).

Line (120-121): *S. aureus* Newman wild-type strain (NWT) imparts more severe arthritis and virulence than its Δ nuc1 mutant during infection in NMRI mice.

We have also added the images of the kidney abscess in the Fig. 1f. This image clearly shows impressively the differences in kidney abscesses between NWT and Δ nuc1 infected mice.

Line 124. Consider removing the word massive

We have replaced massive with significant.

Line 126- 127. The reasoning for this selection of cytokines needs to be introduced.

One could argue for a number of other pro-inflammatory cytokines such as IL-1b, GCSF and GMCSF to be measured in the absence of this reasoning. Mentioning this in the discussion does not aid the interpretation of the results.

Thanks for pointing it out and we have added more introductions.

Line (129-134): IL-6 and TNF- α are essential for septic arthritis development. S100A8/A9 serves as a predictor of septic arthritis in bacteremic mice, and KC (CXCL1) recruits neutrophils, key innate immune cells essential for disease control. Accordingly, on day 7 post-infection we collected blood samples from the infected mice and measured the levels of these immune mediators: IL-6, TNF- α , KC (CXCL1), and S100A8/A9.

Line 148- Virulence

We have corrected it.

Line 158- The reader is not aware of what JE2 is, furthermore the correct annotation for the first time this strain is introduced, is USA300 JE2. (Mentioned later in line 172 and then switches back).

We have corrected it when it is first introduced.

Line (161-172): As staphylococcal macromolecules are frequently contaminated with lipoproteins/lipopeptides that are sensitively detected by Toll-like receptor 2 (TLR2) at picomolar levels, cytokine induction could be due to these constituents. Therefore, the mutant *S. aureus* USA300 LAC JE2 Δ lgt was tested here as a control to recognize a possible interference of TLR2 and TLR9 ligands on cytokine production. JE2 is a plasmid-cured derivative of USA300 LAC strain, that is still methicillin-resistant and an important model strain to study *S. aureus* virulence. JE2 Δ lgt lacks the phosphatidylglycerol:prolipoprotein *diacylglycerol transferase* Lgt, and therefore no lipidation of lipoproteins takes place and no TLR2 response can be triggered by this mutant. By including this mutant and the double mutant JE2 Δ nuc1 Δ lgt together with JE2 and JE2 Δ nuc1 in the comparative immunostimulation studies, it is possible to specifically detect NucA-induced cytokine induction.

Line 172- 173. This is an overstated sentence. There are many studies where USA300 and Newman do not align on virulence properties, and the results depend on the infection context. This should be stated here.

We agree that our previous description about USA300 and Newman was overstated. We have revised the sentence and added the references.

Line (184-190): Comparative virulence studies in different *S. aureus* model strains have shown that the USA300 and Newman strains not only have comparable lethality in a mouse sepsis model but have also similar in their hemolytic activity and biofilm formation⁶. However, in other studies have shown that the two strains exhibit distinct virulence properties; their pathogenicity differs depending on the infection model. Both strains also express NucA and the corresponding Δ nuc1 mutants could be complemented by pRB473-nuc1 (**Supplementary Figure 1d**).

Line 175- 177. It seems like the introduction of the two types of nucleases is relevant at the beginning of the paper where the authors first mention nuc1 as the virulence factor of interest.

We fully agree and have put it into Introduction part.

Line (47-49): Of the two nucleases, it is the NucA that plays the crucial role in degrading extracellular DNA and RNA (eDNA and eRNA, respectively).

Line 199- Typo. Cytokine not cytokines

We have corrected it.

Line 207- This is the first time these cytokines are introduced. The equivalent of KC would be IL-8. IL-10 is not classically pro-inflammatory. The selection of IL-1Ra is

unclear to me. The discussion provides a very detailed explanation of TNF- α and IL-6 but still doesn't tell me about these other cytokines. Additionally, introducing the context at the end of the paper diminishes its value for data interpretation. The revised text in the manuscript just says TNF. That's not a cytokine.

Thank you for your thoughtful thinking. We agree that the choice of immune mediators presented in this section differs from those highlighted in Fig. 2. *In vivo* assay (Fig. 2), we focused on immune mediators that are known to be produced in the septic arthritis model. However, in this particular section for *in vitro* assay, we utilized *in vitro* stimulation of macrophages, osteoblasts, and neutrophils. Consequently, the selection of immune mediators was guided by the molecules most relevant to the activation and response patterns of these cell types under *in vitro* conditions. We acknowledge that IL-10 is anti-inflammatory rather than classically pro-inflammatory. However, its crucial role in regulating and balancing inflammation makes it highly relevant in the context of infection and immune response, as IL-10 levels typically increase during inflammation. Similarly, the selection of IL-1Ra was intended to explore counter-regulatory mechanisms that could influence the inflammatory response. As a natural antagonist of IL-1 signaling, IL-1Ra provides valuable insights into the interplay between pro-inflammatory and anti-inflammatory mediators within this system.

We have expanded our introductions and discussion about cytokines selection and also corrected the TNF to TNF- α .

Line (218-222): To assess the immune responses elicited by different bacterial strains, we measured cytokine production, including classically pro-inflammatory mediators such as TNF- α and IL-6, as well as anti-inflammatory cytokines such as IL-10 and IL-1Ra. IL-10 was also very high in sepsis patients and IL-1Ra and IL-10 also influence the inflammatory response.

Line (365-376): In addition to measuring pro-inflammatory cytokines, we also measured the production of anti-inflammatory cytokines, IL-10 and IL-1Ra, in neutrophils to assess the balance between pro- and anti-inflammatory responses during infection. IL-10 is critical for regulating and balancing the infection and immune response, which typically increase during inflammation. IL-1Ra, as a natural antagonist of IL-1 signaling and related to extensive IL-10 production, could influence the inflammatory response and is used to treat inflammation. IL-1R knock-out mice developed more severe septic arthritis but IL-1R treatment for inflammation-related diseases could also increase the infections in sepsis. When neutrophils were exposed to parental strain and its mutants, JE2 Δ nuc1 showed less IL-10 and IL-1Ra production. Those results give a hint that NucA has an impact on immune stimulation.

Line 212-218. These are very strong assumptions made about the relative virulence differences between Newman and USA300 that are not supported by the data.

Cytokine readouts (especially choosing 3) can be similar for many reasons and can elicit vastly different local and systemic responses during infection. Considering these macrophage and bone cells are not primary and that these phenotypes are not genetically complemented, the language of these statement is too strong and needs to be dialed back.

Thanks for pointing it out. We have now removed it and updated the text.

Line (227-234): It is not unexpected that the Δ lgt mutants induce less cytokines since lipoproteins/lipopeptides trigger a very strong innate immune response. As compared with NWT, there was less IL-6 production in RAW 264.7 and no significant difference in IL-6 and TNF- α in SAOS-2 when exposed to Δ nuc1 strain (**Supplementary**

Figure 3). As the generated mutants showed hardly any difference in growth and hemolytic activity as compared to the parental strain (**Supplementary Figure 1b-d**), we assume that all the effects seen are mainly due to the deletion of *nuc1* or *lgt* genes. Line 231- 233. What is this assumption based on? Literature? Reference required.

We have this assumption based on previous paper but overstating the conclusion. Now, we have revised it to ensure it accurately reflects the existing literature.

Line (246-249): We determined bacterial survival and phagocytosis in neutrophils, following a previously used protocol. With different conditions tested, we found that better results were obtained with an MOI=2 for the phagocytosis assay and an MOI=0.1 for the bacterial killing assay.

Line 244- The complementation was made on an episome?

Yes. The complementary strain was made with a plasmid, pRB473-*nuc1*.

Figure 5 is very confusing. The legend mentions that RFU are plotted in A, but of what, is completely lost. The red anti-citrullinated histone stain is not easy to see, the channels should be separated in order for the red channel to be highlighted. The novelty of these findings is also unclear <https://pmc.ncbi.nlm.nih.gov/articles/PMC2982853/>. Explaining the reasoning for these experiments in the context of this story, at the beginning of this section would assist the manuscript.

We have updated the figure legend 6 to explain what the RFU is.

Line (296-297): Relative fluorescence units (RFU) of Sytox Green normalized to Triton X-100-lysed neutrophils are shown.

We also agree that the red dots are not so clear and have now separated them into different channels. The updated images are shown in Supplementary Figure 5 with DNA, MPO and citH3 channels separated.

We have also added more introductions in this context.

Line (274-278): Neutrophils act as the first defense line of the innate immune system and form NETs to clear pathogens. NETs consist of a DNA backbone coated with various proteins, such as myeloperoxidase (MPO), nuclear proteins (histones), neutrophil elastase (NE), and calprotectin. In *S. aureus*, NucA can degrade extracellular DNA, thereby reducing NET formation and evading immune clearance.

Line 276. Representative images should be demonstrated, even when not exemplary.

We have updated representative images in Fig. 6b.

Line 285- The word massive is not scientific or specific and should not be used to describe data.

We have replaced the massive to marked, severe or pronounced.

Line 310- 318. If TLR-9 is the only thing that initiates a host response to *S. aureus* please provide references of this statement. This entire paragraph has no references.

Not only the TLR9 pathway but also other pathways could induce septic arthritis. We have now added relevant references to state and ensure the entire paragraph is well-documented.

Line 330- 336. Are there any references for this paragraph?

We have included the reference and updated the text.

Line (353-356): As we showed that NucA-digested bacterial eDNA has no immune-stimulating activity in murine macrophages, we expected that the $\Delta nuc1$ mutant, in which eDNA is not degraded (**Supplementary Figure 1a**), would elicit a stronger immune response than the parental strain in various cell types.

Line 527- Typo. Neutrophil not neutrophils

We have corrected it.

Supplement lines 28-29. Grammar

We have corrected it.